# Extreme Saharan-dust events expand northward over the Atlantic and Europe prompting record-breaking PM$_{10}$ and PM$_{2.5}$ episodes

Sergio Rodríguez[1,*], Jessica López-Darias[1]

[1]Consejo Superior de Investigaciones Científicas, IPNA CSIC, Tenerife, Canary Islands, Spain.

*Correspondence to*: Sergio Rodríguez (sergio.rodriguez@csic.es)

**Abstract.** Unprecedented extreme Saharan-dust (duxt) events have recently expanded northward, from subtropical NW Africa to the Atlantic and Europe, with severe impacts on the Canary Islands, mainland Spain and continental Portugal. These six historic duxt episodes occurred on 3-5 February 2020, 22-29 February 2020, 15-21 February 2021, 14-17 January 2022, 29 January - 1 February 2022 and 14–20 March 2022. We analysed data of 341 Governmental Air Quality Monitoring Stations

(AQMS) of Spain (330) and Portugal (11), where PM$_{10}$ and PM$_{2.5}$ are measured with European EN-standards, and found that during duxt events PM$_{10}$ concentrations are underestimated due to technical limitations of some PM$_{10}$ monitors to properly measure extremely high concentrations. We assessed the consistency of PM$_{10}$ and PM$_{2.5}$ data and reconstructed 1690 PM$_{10}$ (1h average) data of 48 and 7 AQMS of Spain and Portugal, respectively, by using our novel *duxt-r* method. During duxt events, 1-hour average PM$_{10}$ and PM$_{2.5}$ concentrations were within the range 1000-6000 μg/m$^3$ and 400-1200 μg/m$^3$, respectively. The

intense winds leading to massive dust plumes occurred within meteorological dipoles formed by a blocking anticyclone over western Europe and a cut-off low located at the southwest, near the Canary Islands, Cape Verde or into the Sahara. These cyclones reached this region by two main paths: deviated southward from the Atlantic mid-latitude westerly circulation or deviated northward from the tropical belt. The analysis of the 2000-2022 PM$_{10}$ and PM$_{2.5}$ time series shows that these events have no precedent in this region. The 22-29 February 2020 event led to (24h average) PM$_{10}$ and PM$_{2.5}$ concentrations within

the range 600-1840 μg/m$^3$ and 200-404 μg/m$^3$, respectively, being the most intense dust episode ever recorded in the Canary Islands. The 14–20 March 2022 event led to (24h average) PM$_{10}$ and PM$_{2.5}$ values within the range 500-3070 μg/m$^3$ and 100-690 μg/m$^3$ in south-eastern, 200-1000 μg/m$^3$ and 60-260 μg/m$^3$ in central and 150-500 μg/m$^3$ and 75-130 μg/m$^3$ in northern regions of mainland Spain and within the ranges 200-650 μg/m$^3$ and 30-70 μg/m$^3$ in continental Portugal, respectively, being the most intense dust episode ever recorded in these regions. All duxt events occurred during northern-hemisphere

meteorological anomalies characterised by subtropical anticyclones shifted to higher latitudes, anomalous low pressures expanding beyond the tropical belt and amplified mid-latitude Rossby-waves. New studies have reported on recent record beating PM$_{10}$ and PM$_{2.5}$ episodes linked to dipoles induced extreme dust events from North Africa and Asia, in a paradoxical context of multidecadal decrease of dust emissions, a topic that requires further investigations.

## 1 Introduction

Airborne dust aerosol particles are a key component of the Earth System influencing on climate (Kok et al., 2023), ecosystems (Yu et al., 2015), fisheries (Rodríguez et al., 2023) and human health (Domínguez-Rodríguez et al., 2021; Tong et al., 2023). Major dust sources are located in North Africa, the Middle East and inner Asia (Prospero et al., 2002), accounting for $\approx$ 75% of global emissions; secondary sources are located in northern and southern America, southern Africa, Australia and at high latitudes (Kok et al., 2023). Because these sources are located in arid regions they have usually been considered as "natural dust-sources" (e.g. the bed of naturally dried ancient lakes (Ginoux et al., 2012; Prospero et al., 2002)); however a growing body of evidences is showing that human actions such as the soil disruption by traditional grazing and agriculture (Katra, 2020; Mulitza et al., 2010; Vukovic et al., 2021), mining (Rodríguez et al., 2011; Zafra-Pérez et al., 2023), drying of water courses and lakes (Ginoux et al., 2012; Govarchin-Ghale et al., 2021), the expansion of intensive agriculture (Lambert et al., 2020) and wildfires (Yu and Ginoux, 2022) are contributing to increase dust emissions.

Dust storms cause huge socio-economic impacts linked to loose of visibility, road traffic disruption and accidents, deviation of air travel or closure of maritime and air navigation space, cardiovascular and respiratory diseases, loose of soil and a decrease in solar energy production (Cañadillas-Ramallo et al., 2022; Domínguez-Rodríguez et al., 2021; Middleton et al., 2021; Miri and Middleton, 2022; Pi et al., 2020). For this reason, a set of operational "dust services" are available to forecast and monitor dust activity by modelling and satellite observations (Mona et al., 2023). In-situ concentrations of $PM_{10}$ and $PM_{2.5}$ (respirable particulate matter -PM- smaller than 10 and 2.5 microns, respectively) regularly measured in air quality monitoring networks are commonly used to assess dust impacts and to validate dust models (Mona et al., 2023). In southern Europe, Saharan dust events tend to increase $PM_{10}$ concentrations up to typical values within the range 40-90 $\mu g/m^3$ (24h average values), while dust events with (24h average) $PM_{10}$ >100 $\mu g/m^3$ are unusual (Millán-Martínez et al., 2021; Pey et al., 2013).

Understanding how climate change is affecting dust emissions is a challenge as these emissions are also affected by the natural atmospheric variability (as traced by ENSO, NAO, AMO and other climatic indexes (Evan et al., 2016)), as well as by the changes in atmospheric circulation and soil properties (e.g. humidity and biological crust (Rodriguez-Caballero et al., 2022)) as the atmosphere warms due to the increasing concentrations of greenhouse gases.

Current climate models are unable to reproduce the historical increase in atmospheric dust loads observed in paleorecords (Kutuzov et al., 2019; Preunkert et al., 2019). Based on models constraining dust emissions, it has been estimated that the global dust mass load in the modern climate is $\approx$ 56% higher than in pre-industrial times (Kok et al., 2023), with a maximum dust load in the mid-1980s and a subsequent decrease attributed to a decrease in North African and Asian dust emissions linked to a slowdown of the atmospheric circulation interconnected to global warming (Evan et al., 2016; Jiang et al., 2023; Liu et

al., 2020; Middleton, 2019; Ridley et al., 2014; Xie et al., 2023). In this scenario of decreasing dust trend in North Africa and Asia, a series of unexpected extreme dust-events have recently occurred.

In March 2018 a "*record-breaking Saharan dust plume*" crossed the Eastern Mediterranean (Kaskaoutis et al., 2019), leading to $PM_{10}$ values of up to (1h-average) 6000 µg/m$^3$ (Solomos et al., 2018), a three-fold increase in hospital admissions (Lorentzou et al., 2019; Monteiro et al., 2022) and an accelerated snow melting in the Caucasus (Dumont et al., 2020). In June 2020 the so-called "*Godzilla record-breaking trans-Atlantic African dust plume*" (Bi et al., 2023; Francis et al., 2020, 2022; Pu and Jin, 2021) led to (24h average) values $PM_{10}$ = 453 µg/m$^3$ in the Caribbean and $PM_{10}$ and $PM_{2.5}$ values = 135 and 74 µg/m$^3$ in southern United States (Yu et al., 2021), respectively. In March 2021, two "*record-breaking dust events*" in China (Gui et al., 2022) led to (1h-average) $PM_{10}$ and $PM_{2.5}$ values of up to 7525 µg/m$^3$ and 685 µg/m$^3$, respectively (Filonchyk and Peterson, 2022; Zhang et al., 2023). In November 2021, an "*extreme dust storm*" in Uzbekistan led to (1h average) $PM_{10}$ and $PM_{2.5}$ concentrations of up to 4575 µg/m$^3$ and 705 µg/m$^3$ (Nishonov et al., 2023; Xi et al., 2023) .

In this study we present a set of unprecedented extreme dust-events that have recently (2020-2022) expanded northward, from North Africa, to the Atlantic and Europe prompting record-breaking $PM_{10}$ and $PM_{2.5}$ episodes in Spain. The observed increase in dust activity in the Western Euro-Mediterranean region has recently been studied based on meteorological modelling reanalysis and aerosol optical depth - satellite measurements (Cuevas et al., 2023). We also focused on the analysis of the consistency of $PM_{10}$ and $PM_{2.5}$ data in the Governmental Air Quality Monitoring Networks during the extreme dust events due to the importance of having suitable data in the public databases used for health effects studies, model validation and constrains etc… (Mona et al., 2023). Understanding these extreme dust events is crucial for this region, since climate projections forecast the expansion of the North African drylands toward the northwest, increasing the risk of desertification of Spain and Portugal as the subtropical anticyclones expand in a warming climate (Cresswell-Clay et al., 2022; Guiot and Cramer, 2016) with an associated increase in the desert dust load (Gomez et al., 2023; Liu et al., 2024).

## 2 Methodology

### 2.1 Data of $PM_{10}$ and $PM_{2.5}$

We analysed the 2000-2022 data of $PM_{10}$ and $PM_{2.5}$ recorded in the Governmental Air Quality Monitoring Network of Spain and Portugal. The data from Spain were recorded in 330 air quality stations distributed across the 17 Autonomous Regions and the Autonomous city of Ceuta, whereas the data from Portugal were collected in 11 stations distributed across Norte, Centro, Lisboa, Vale do Tejo, Alentejo, Algarve, Madeira and Azores regions.

These stations are integrated in the European Air Quality Monitoring Network, which is the largest European infrastructure for $PM_{10}$ and $PM_{2.5}$ monitoring following standardized methods for measurements, quality assurance (QA) and quality control (QC) (EN-16450:2017 and EN-12341:2015). In these stations, high temporal (10 to 60 minutes) resolution $PM_{10}$ and $PM_{2.5}$

data are obtained using automatic monitors based on different principle of measurement, such as beta attenuation, tapered element oscillating microbalance and optical particle sizers (Rodríguez et al., 2012), with technical specifications accomplishing the EN-16450:2017 standard. Data of 24h average $PM_{10}$ and $PM_{2.5}$ are also obtained with the gravimetric reference method (EN-12341:2015), used for QA/QC assessments and for converting the $PM_{10}$ and $PM_{2.5}$ data obtained with the automatic devices to gravimetric equivalent data by using the European reference protocols (EN-16450:2017). The $PM_{10}$ and $PM_{2.5}$ data that we used were provided by the Ministry of Ecological Transition and Demographic Challenge and by the air quality departments of the Autonomous Regions of Spain and the Agência Portuguesa do Ambiente.

## 2.2 Complementary modelling and satellite data

For each specific study event, we also used data of the National Center for Environmental Prediction/National Center for Atmospheric Research (NCEP/NCAR) meteorological reanalysis (Kalnay et al., 1996), of the Modern-Era Retrospective analysis for Research and Applications, Version 2 (MERRA-2) model dust reanalysis (Gelaro et al., 2017) and of satellite Visible Infrared Imaging Radiometer Suite (VIIRS) images sensor onboard the Suomi polar satellite.

## 3 Results and discussion

The set of extreme dust events, to which we will refer with the acronyms *duxt* or *dx* episodes, occurred on 3-5 February 2020 (dx-01), 22-29 February 2020 (dx-02), 15-21 February 2021 (dx-03), 14-17 January 2022 (dx-04), 29 January - 1 February 2022 (dx-05), and 14–20 March 2022 (dx-06). These duxt events were characterised by dark and reddish "apocalyptic" skies (Fig.1A-1C). National Spanish and international media (Fig.1), the European Copernicus (Fig.1D) and NASA - Earth Observatory (Fig.1E) platforms reported on these historic events and their impacts on socio economical activities. During the impact of the duxt-02 event in the Canary Islands, record-breaking temperatures occurred, wildfires were favoured by the windy and dry conditions, solar production energy dropped by a 70%, maritime and air navigation space was closed, and thousands of flights were cancelled with huge economic implications linked to the transfer of tourists between the Canary Islands and Europe (Fig.1)(Cuevas et al., 2021).

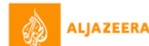

**Thousands evacuated as Canary Isles sandstorm fuels wildfires** 24 Feb 2020

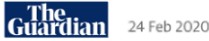 24 Feb 2020

**Tourists stranded in Canary Islands after Saharan sandstorm blows in**

**Dozens of flights cancelled due to poor visibility, leaving holidaymakers stuck at airports**

**EL PAÍS**

**La calima, el viento y el fuego cierran el espacio aéreo y marítimo en Canarias** 24 FEB 2020 ·

**El episodio de calima de Canarias bate récords de temperatura** 24 FEB 2020 ·

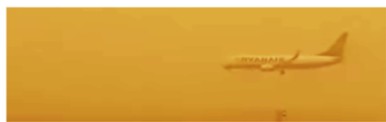

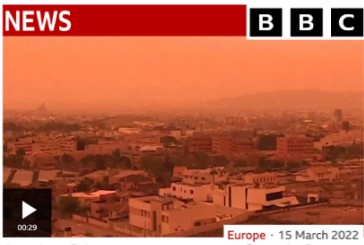

**Spain skies turn orange after Saharan dust cloud sweeps over country** Europe · 15 March 2022

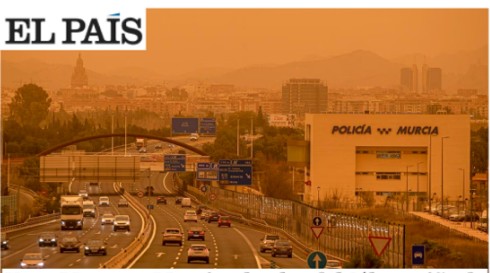

**No es Marte, es Murcia: el polvo del Sáhara tiñe de rojo el cielo de España** 15 MAR 2022 - 11:32 CET

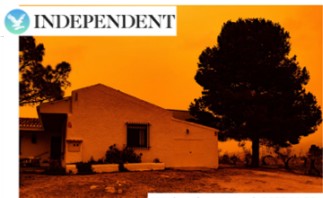

Wednesday 16 March 2022 16:38

**The Washington Post** February 24, 2020

**Canary Islands sandstorm grounds flights, closes schools as Sahara dust moves into open Atlantic**

**Dust from Sahara turns sky in Spain to 'Bladerunner' orange**

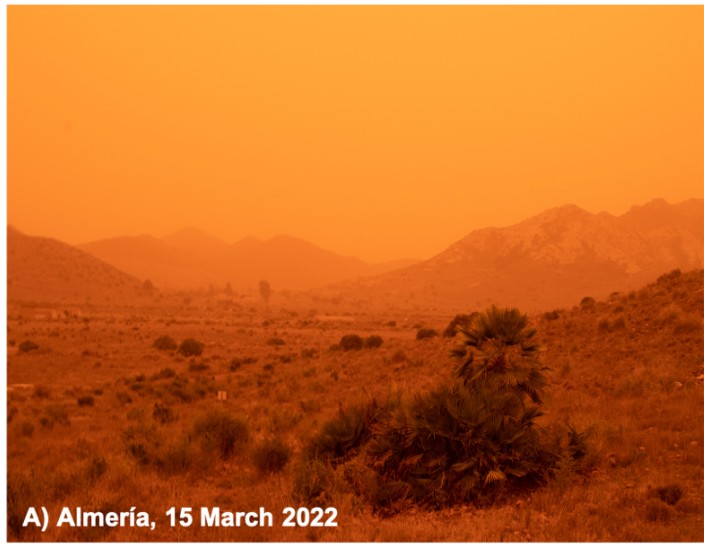

A) Almería, 15 March 2022

**Figure 1. News on the extreme dust events impacting mainland Spain and the Canary Islands published in international and Spanish national media. Picture of the Cabo de Gata in south-eastern mainland Spain (Almería province) the 15 of March 2022 (dx-06) (taken and provided by Eva de Mas Castroverde).**

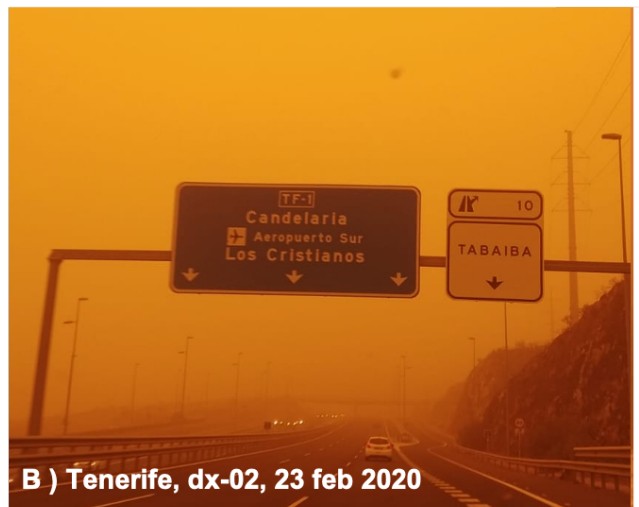

**B ) Tenerife, dx-02, 23 feb 2020**

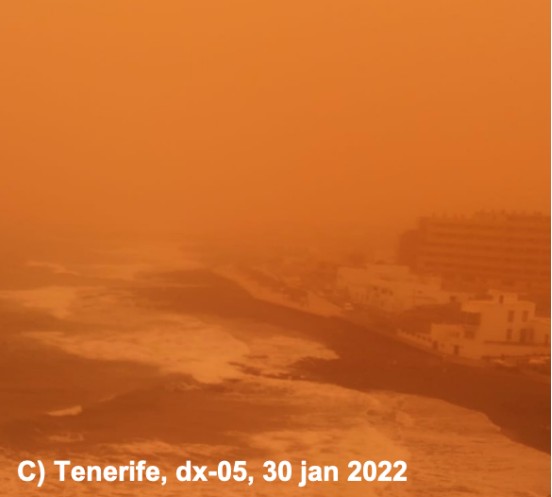

**C) Tenerife, dx-05, 30 jan 2022**

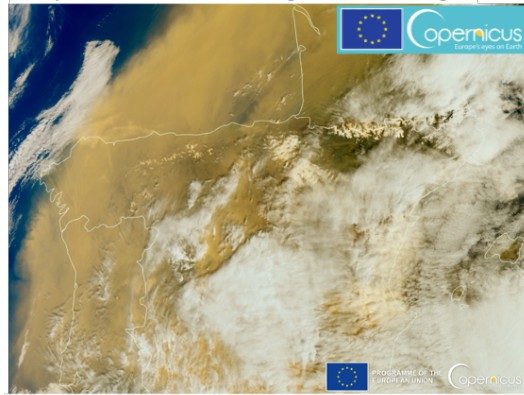

**D )** "Historic" Saharan dust episode in western Europe – CAMS predictions accurate

15th March 2022

*Southwestern Europe is currently experiencing an exceptional Saharan dust episode, which has been turning skies across the region*

*Sentinel-3 visible imagery showing Saharan dust over southwest Europe on 15 March 2022.*

Source: European Union, Copernicus Sentinel-3 imagery

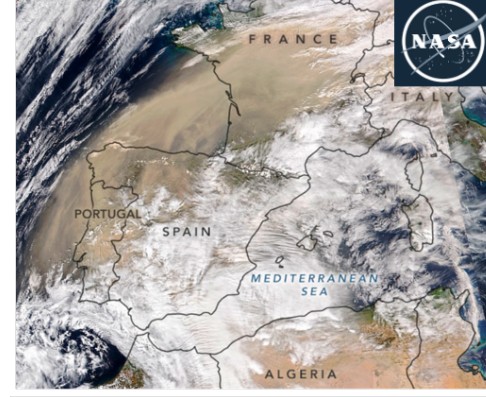

**E )** An Atmospheric River of Dust

March 15, 2022

On March 15, 2022, a plume of Saharan dust was blown out of North Africa and across the Mediterranean into Western Europe. The dust turned skies orange, blanketed cities, impaired air quality, and stained ski slopes.

The image above, acquired on March 15 by the Visible Infrared Imaging Radiometer Suite (VIIRS) on the NOAA-20 spacecraft, shows the dust plume moving out of Algeria and over the Iberian Peninsula.

**Figure 1 (continue). Pictures of Tenerife (Canary Islands) the 23 of February 2020 (dx-02) and 30 of January 2022 (dx-05) (taken by the authors). Composite of the websites of Copernicus (https://atmosphere.copernicus.eu/historical-saharan-dust-episode-western-europe-cams-predictions-accurate) and Earth Observatory (https://earthobservatory.nasa.gov/images/149588/an-atmospheric-river-of-dust) reporting on the historic dust event the 15th of March 2022.**

### 3.1 Assessment and reconstruction of PM$_x$ data during the duxt events

We analysed the data of PM$_{10}$ and PM$_{2.5}$ (PM$_x$) of 330 Air Quality Monitoring Stations (AQMS) of Spain and of of Portugal. We found that during the duxt events, the ½ hour and 1 hour resolution data of PM$_{10}$ increased up to reach a rather

constant 'saturation' value, that in most of cases was somewhat lower than 1000 µg/m$^3$ (in many cases between 900 and 1000 µg/m$^3$), no values above this threshold appear in the data records. In these cases, PM$_{10}$ concentrations remained constant during the period of extremely high dust concentrations (typically 5 to 30 hours), a behaviour that was not generally observed in PM$_{2.5}$, which exhibited a regular variability (with values < 1000 µg/m$^3$) and even increases in the periods when PM$_{10}$ remained (un-consistently) constant. This behaviour can be observed in the time series of PM$_{10}$ (Fig.2A1-A3) and PM$_{2.5}$ (Fig.2B1-B3)

linked to the dx-01, dx-02, dx-04, dx-05 and dx-06 events (dx-03 is not included in Fig.2 for the sake of brevity). This saturation threshold close to ≈1000 µg/m$^3$ is the upper operation limit of some PM$_{10}$ monitors and is also the top value of the validation data flag in some data recording commercial software used in many governmental air quality monitoring networks, which assume that PM$_{10}$ concentrations above this threshold may suffer underestimation due to the high load of particles (e.g. accumulated in the filter-tape of the beta instrument or particle coincidence problems in the optical particle sizers leading to a

loss of sensitivity) and consequently do not record values above this threshold. In some AQMS the saturation threshold was found at 500 µg/m$^3$ and even at 200 µg/m$^3$. In fact, the EN-16450, EN-12341 and EN-14907 accreditations for PM$_{10}$ and/or PM$_{2.5}$ monitors available in the European market are obtained for specific ranges, whose most frequent upper limit is 1000 µg/m$^3$ for many monitors (e.g. Met One™ BAM-1020, Comde Derenda™ APM 2 and Thermo Fisher Scientific™ 5014i, 5030i SHARP, TEOM 1405-F and 1405-DF), although is as low as 200 µg/m$^3$ for some equipment's (e.g. FAI™ Swam 5a)

and is, in contrast, as high as 10000 µg/m$^3$ for other devices (e.g. PALAS™ Fidas 200 and 200E) according to the certifications agencies (e.g. TÜV, see https://www.qal1.de/). In all these cases of PM$_{10}$ data affected by saturation, we reconstructed the PM$_{10}$ concentrations with the method described in Fig. 3, which we have called *duxt-r* "PM$_x$ evaluation and reconstruction method based on ratios during extreme dust events".

In this *duxt-r* method: 1) we first identified the invalid 1 hour (or ½ hour or 10 minutes time resolution) PM$_{10}$ data affected by

saturation and the associated invalid PM$_{2.5}$/PM$_{10}$ ratios, highlighted with red circles in the example of the dx-04 event shown in Fig. 3A and 3B, 2) then, the PM$_{2.5}$/PM$_{10}$ ratio during the PM$_{10}$ saturation period was estimated by linear interpolation between the last valid PM$_{2.5}$/PM$_{10}$ data before saturation and the first valid PM$_{2.5}$/PM$_{10}$ after saturation [R$_{2.5/10}$(i)], highlighted with green points in Fig. 3B, 3); as results of this interpolation the PM$_{2.5}$/PM$_{10}$ ratios we used is not constant, it changes hour by hour, e.g. from 0.184 to 0.162 in the example shown in Fig. 3B. Finally, the 1-hour (or ½ hour or 10 minutes) resolution

PM$_{10}$ concentrations were determined with equation-01. The method was validated with comparison with data recorded in a few PM$_{10}$ monitors not affected by saturation (described below).

$$PM_{10} = \frac{PM_{2.5}}{R_{2.5/10}(i)} \tag{1}$$

We found that the PM$_{2.5}$/PM$_{10}$ ratio during the duxt events were within the range 0.16 to 0.22 in most of AQMS, a value lower than that observed during regular dust events (typically ≈0.3) and much lower than that observed in environments affected by secondary particle formation and by vehicle exhaust and other combustion emissions (typically within 0.6-0.9) (Rodríguez and López-Darias, 2021).

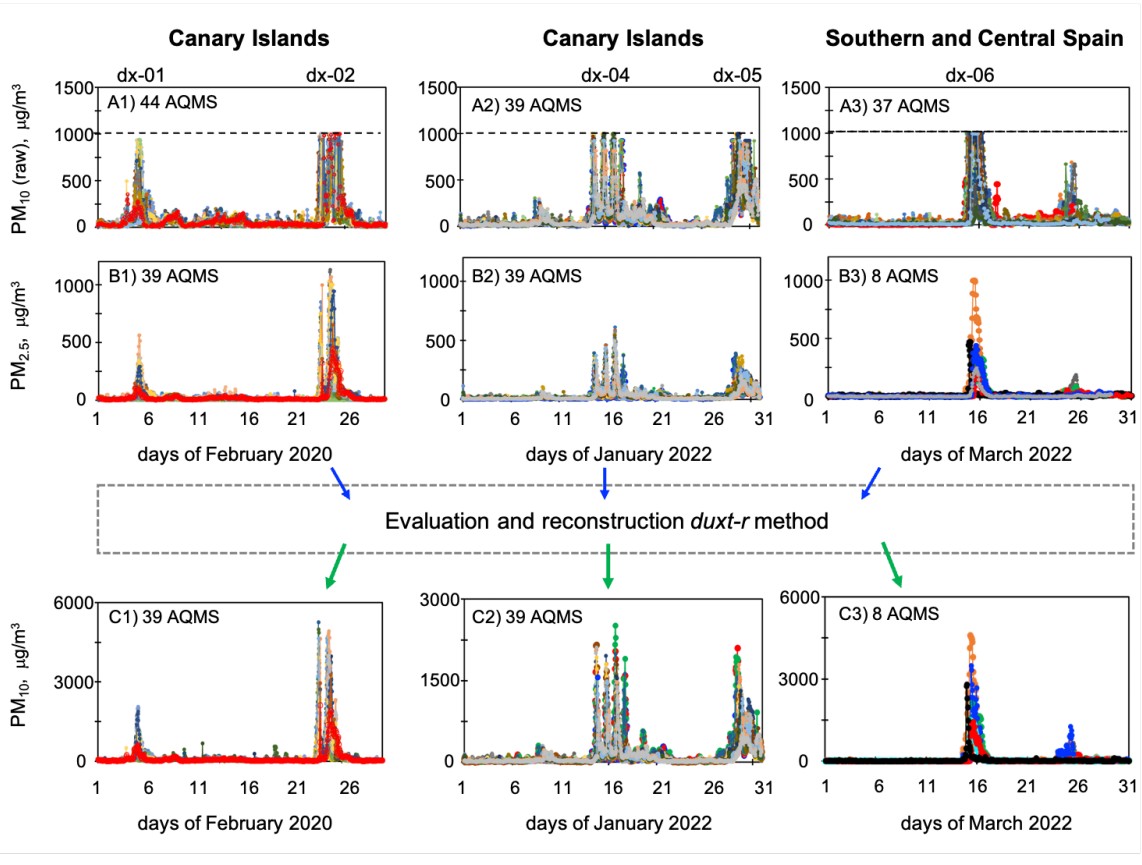

**Figure. 2. Time series of 1h average PM$_{10}$ (raw) (A1-A3) and PM$_{2.5}$ (B1-B) and evaluated and reconstructed PM$_{10}$ data (C1-C3) during February 2020, January 2022 and March 2022, indicating the duxt events. The number of AQMS included in each plot is shown. PM$_{10}$ (raw) means original raw (non-reconstructed) data (A1-A3). Data of PM$_{2.5}$ are also raw (B1-B3). PM$_{10}$ data of plots C1-C3 combines measured valid and reconstructed data.**

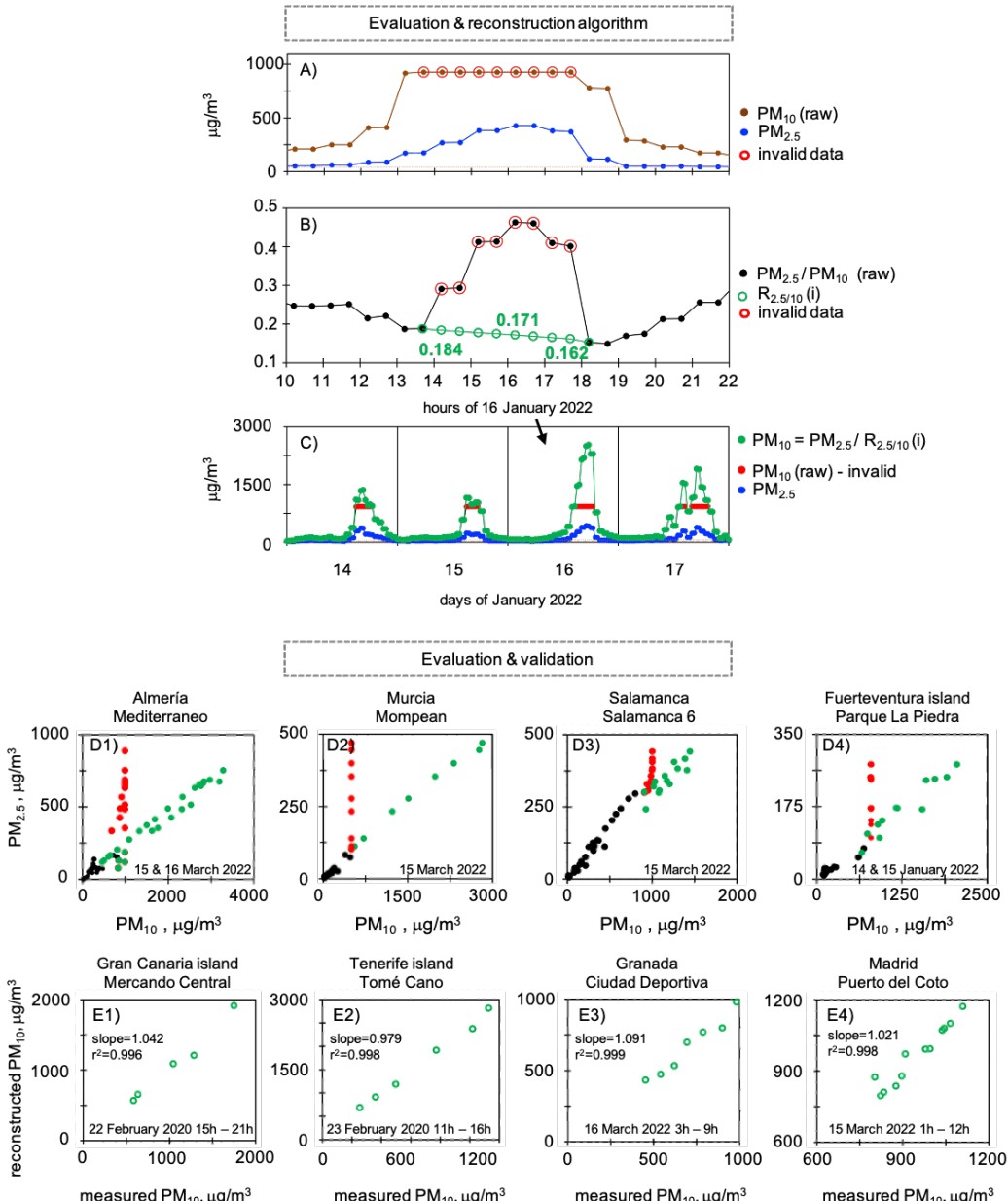

**Figure 3. Evaluation, reconstruction and validation of data with the duxt-r method. Plots (A-C) shows data of El Charco site (Fuerteventura) during dx-04 event (14-17 January 2022): (A) data of PM$_{10}$ (raw, i.e. including saturated values) and PM$_{2.5}$, (B) the measured PM$_{2.5}$/PM$_{10}$ (raw) ratio and the interpolated PM$_{2.5}$/PM$_{10}$ ratio [R$_{2.5/10}$(i)], and (C) PM$_{2.5}$, PM$_{10}$ (raw) and PM$_{10}$ reconstructed data. Invalid data due to PM$_{10}$ underestimation are highlighted with red circle (A and B) and red points (C and D). D1-D4) scatterplot of PM$_{2.5}$ and PM$_{10}$ data highlighting valid data (black circle), invalid PM$_{10}$ data (red circle) and reconstructed PM$_{10}$ data (green circle). E1-E4) reconstructed versus measured PM$_{10}$ data. Green number in plot C indicates some of the values of the PM$_{2.5}$/PM$_{10}$ interpolation.**

In our database, we replaced the $PM_{10}$ saturated data by the new $PM_{10}$ re-constructed data, i.e. the red (saturated) points shown in Fig. 3C were replaced by the green re-constructed points. At each site (AQMS), the consistency of the reconstructed data was assessed by analysing the scatter plot of the $PM_{2.5}$ versus $PM_{10}$ data (Fig. 3D1-3D4). As example, the results obtained in AQMS located in Almería province (Mediterraneo AQMS) and Murcia (Mompean AQMS) in south-eastern Spain, Salamanca province (Salamanca-6 AQMS) in central northern Spain, and in Fuerteventura (Canary Islands) (Fig. 3D1-3D4) is shown (Fig.3D1-3D4). With this method, the red $PM_{10}$-saturated data shown in Fig. 3D1-3D4 were replaced by the green (reconstructed) data.

Because of the technical manufacturing specification, the automatic $PM_{10}$ and $PM_{2.5}$ monitors of two AQMS were able to record valid and consistent $PM_{10}$ and $PM_{2.5}$ data higher than 1000 $\mu g/m^3$. We used these records to validate this methodology (Fig.3E1, 3E2 and 3E4). At these sites, we re-constructed the $PM_{10}$ concentrations above 1000 $\mu g/m^3$ as if they had experienced saturation and then we compared the "reconstructed versus the measured $PM_{10}$ concentrations"; for this comparison we also included data between 500 and 1000 $\mu g/m^3$ for the purpose of having a larger dataset (i.e. all $PM_{10}$ data > 500 $\mu g/m^3$). We found that the difference between re-constructed and measured $PM_{10}$ concentrations ranged between 2 and 9% (Fig.3E1-3E4). In other few AQMS using beta attenuation devices able to provide $PM_{10}$ > 1000 $\mu g/m^3$, we found a low $PM_{10}$ variability above this threshold (indicating loose of sensitivity due to mass overload in the filter tape) that was inconsistent with the variability in $PM_{2.5}$ and that resulted in $PM_{2.5}/PM_{10}$ ratios similar to those affected by the saturation as described above (red circles in Fig.3A); at these sites we also reconstructed the $PM_{10}$ with the *duxt-r* method (Fig.3). Finally, the $PM_{10}$ data measured with the automatic monitors (measured and reconstructed) were converted to gravimetric equivalent data by intercomparison with $PM_{10}$ data obtained with the gravimetric reference method (EN-12341:2015; 24h sampling, available during 7 to 25 days/month depending on the AQMS) using the standardised procedure (EN-16450:2017).

The new dataset obtained with this *duxt-r* method ($PM_{10}$ measured and reconstructed and then converted to gravimetric equivalent) evidences that 1-hour average $PM_{10}$ data that appeared as "saturated" at 1000 $\mu g/m^3$ actually reached values close to 6000 $\mu g/m^3$ in the dx-02 event, close to 1400 $\mu g/m^3$ in the event dx-03, close to 2000 $\mu g/m^3$ in the events dx-01, dx-04 and dx-05, and between 3000 and 4500 $\mu g/m^3$ in the events dx-06 (Fig.2C1-2C3).

By applying this methodology, we reconstructed a total of 1690 hourly $PM_{10}$ data: 1537 hourly $PM_{10}$ data belonged to (i) 48 AQMS of Spain, distributed between the regions Canary Islands (39), Andalucía (5), Murcia (1) and Castilla y León (2) and Madrid (2) and (ii) 153 hourly $PM_{10}$ data belonged to 7 AQMS of Portugal, distributed between the regions Lisboa – Vale do Tejo (6) and Alentejo (1). The data we reconstructed with this method are already available in public data bases of the Governmental Air Quality Networks of Spain, the Ministry of Ecological Transition and the European Environment Agency. The $PM_{10}$ data of other 44 AQMS that also experienced $PM_{10}$ saturation could not be reconstructed due to the lack of

simultaneous $PM_{2.5}$ measurements, a total of 655 hourly data in these AQMS located in the Canary Islands (14), Andalucía (10), Extremadura (2), Castilla y León (11) and Murcia (7). Just to illustrate the huge importance of re-constructing the data, a brief comparison (for a few AQMS) of the 24h average $PM_{10}$ concentrations calculated with saturated $PM_{10}$ data vs reconstructed $PM_{10}$ data: (1) 948 vs 3069 µg/m$^3$ 15 March 2022 in Almería province (Mediterraneo AQMS), (2) 740 vs 1840 µg/m$^3$ 23 February 2020 in Gran Canaria (Playa del Inglés AQMS), (3) 1238 vs 1684 µg/m$^3$ 23 February 2020 in Tenerife (Tomé Cano AQMS), (4) 577 vs 1421 µg/m$^3$ 23 February 2020 in Tenerife (Piscina Municipal AQMS), and (5) 527 vs 621 µg/m$^3$ 15 March 2022 in Granada province (Palacio de Congresos AQMS). The maximum 1-hour average $PM_{10}$ and $PM_{2.5}$ recorded during dx-01 to dx-06 are in a selection of AQMS is shown in Fig.S1 and S2.

## 3.2 Analysis of the extreme dust events

The first two events occurred in February 2020 (Fig.4): 3-5 February 2020 (dx-01; Fig.4A1) and 22-29 February 2020 (dx-02; Fig.4A2). Throughout six weeks (from mid-January to ending February) a blocking anticyclone established over Iberia, i.e. the Iberian Peninsula (Fig. 4F and 4H), resulting in anomalous easterly wind over central Algeria (wind anomaly not shown for the sake of brevity), a scenario favourable to dust events (Alonso-Pérez et al., 2011a). On 3-5 February 2020 a cyclone reached Cape Verde, the low (over Cabo Verde) to high (over Iberia) L-to-H dipole configuration (Fig.4F) resulted in a strong pressure/geopotential gradient and winds that prompted dust emissions and a dense plume of Saharan dust that expanded over the Atlantic to the Canary Islands and toward the Azores (Fig.4G). Across the Canary Islands the dx-01 event resulted in (i) 1h average $PM_{10}$ and $PM_{2.5}$ concentrations within the range 300-2100 µg/m$^3$ (Fig.2C1) and 100-400 µg/m$^3$ (Fig.2B1; Fig.S1A1-A2), respectively, and (ii) 24h average $PM_{10}$ and $PM_{2.5}$ concentrations within the range 100-535$^x$ µg/m$^3$ (Fig.4A1; Fig.5A1; $^x$ = maximum at Tenerife, San Miguel Tajao AQMS) and 50-165$^x$ µg/m$^3$ (Fig.4A2; Fig.5A2) ($^x$Tenerife, Tomé Cano AQMS), respectively. In Madeira the dust impact was smoother, with 24h average $PM_{10}$ concentrations within the range 50-115 µg/m$^3$ (Fig.4C), due to this island remained aside the core of the dust plume (Fig.4G). Dust events prompted by the summer North African dipole (NAFDI) were originally introduced by Rodríguez et al. (2015). This concept of meteorological L-to-H dipoles has also been found to drive dust events in the Middle East (Kaskaoutis et al., 2015, 2017) and the June 2020 Godzilla duxt event (Francis et al., 2020).

On 22 February 2020, a new cyclone reached again the region of Cape Verde, resulting in a similar L-to-H dipole meteorology (Fig.4H) which prompted a dusty jet stream in the subtropical North Atlantic, impacting the Canary Islands (Fig.4A2, 4H and 4I). This scenario caused the dark oranges skies, record breaking temperatures, wildfires linked to strong dry winds, closure of maritime and air navigation space and the massive (thousands) flight cancellations described above (Fig.1)(Cuevas et al., 2021). During this dx-02 event, extreme $PM_x$ concentrations were recorded in the Canary Islands, with: (i) 1h average $PM_{10}$ and $PM_{2.5}$ concentrations within the range 2000-5254$^x$ µg/m$^3$ (Fig.2C1; Fig.S1B1-B2) ($^x$Gran Canaria, Arinaga AQMS) and

400-1129[x] µg/m$^3$ ($^x$Gran Canaria, Mercado Central AQMS) (Fig.2B1; Fig.S1), respectively, and (ii) 24h average PM$_{10}$ and PM$_{2.5}$ concentrations within the range 600-1840[x] µg/m$^3$ (Fig.4B1, 5B1) ($^x$Gran Canaria, Playa del Inglés AQMS) and 200-404[x] µg/m$^3$ (Fig.4B2, 5B2) ($^x$Gran Canaria, Playa del Inglés AQMS), respectively. Concentrations (24h average) of PM$_{10}$ during this dx-02 event were up to 6 times higher than the upper limit of PM$_{10}$ during the regular dust events in the Canary Islands

($\approx$300 µg/m$^3$) and also much higher than the extraordinary 680 µg/m$^3$ recorded during the dust event of 26 January 2000 (Viana et al., 2002). During this dx-02 episode, the highest 1-hour PM$_{10}$ (3500-5254 µg/m$^3$) and PM$_{2.5}$ (800-1129 µg/m$^3$) and 24-h average PM$_{10}$ (1200-1840 µg/m$^3$) and PM$_{2.5}$ (230-404 µg/m$^3$) concentrations were recorded in the AQMS located in the central part of the dust plume, in Gran Canaria and Tenerife islands (Fig.5B1-5B2)(Fig.S1). After the 24$^{th}$ February 2020, the Saharan dust plume shifted northward over the Atlantic reaching mainland Spain (Fig.4K), resulting in (24h average) PM$_{10}$

concentrations within the range 70-155 µg/m$^3$ (Fig.4D1) in central Spain (Madrid, Extremadura and Castilla La Mancha regions), 70-75 µg/m$^3$ (Fig.4D1) in central Portugal and 80-200 µg/m$^3$ in eastern Spain (Comunidad Valenciana region) (Fig.4E), and PM$_{2.5}$ concentrations within the range 20-33 µg/m$^3$ in central Spain and central Portugal (Fig.4D2), respectively. The re-analysis of MERRA-2 properly tracked these two dx-01 & dx-02 events, with dust and dust$_{2.5}$ (i.e. dust in the PM$_{2.5}$ fraction) within the range of the PM$_{10}$ and PM$_{2.5}$ concentrations recorded in the AQMS (see orange circles in Figure 4B-4D).

The third duxt event (dx-03: 15-21 Feb 2021) (Fig.6D) was also caused by the intense wind (Fig.6C) linked a dipole L-to-H meteorology (Fig.6B), with the associated blocking anticyclone located over Iberia and a cyclone over the Sahel (Fig.6B). The impact on the Canary Islands occurred during 15-19 February (Fig.6A1-A2) and in Madeira during 16-18 February 2021 (Fig.6A3). In the Canaries, the highest 1-hour average PM$_{10}$ (1000-1352 µg/m$^3$) and PM$_{2.5}$ (200-326 µg/m$^3$) were recorded in Gran Canaria, Tenerife, La Gomera and La Palma (Fig.S1C1-C2). The 16 February 2021 resulted in 24h average PM$_{10}$ and

PM$_{2.5}$ concentrations within the range 400-711[x] µg/m$^3$ (Fig.5; Fig.6A1)($^x$Las Galletas AQMS, Tenerife) and 80-205[x] µg/m$^3$ (Fig.5; Fig.6A2)($^x$Las Galanas, La Gomera), respectively. Subsequently, the dusty air mass tracked the northward anticyclonic circulation, resulting in (24h average) PM$_{10}$ concentrations within 80-180 µg/m$^3$ in Madeira (Fig. 6A3), reaching central mainland Portugal and Spain (18-21 Feb 2021; Fig.6E), and resulting in 24h average PM$_{10}$ and PM$_{2.5}$ concentrations within the range 75-150 µg/m$^3$ (Fig.6A4) and 20-55 µg/m$^3$ (Fig. 6A4-6A5), respectively. The MERRA-2 reanalysis properly tracked

the dx-03 event, except during 16-17 of February in Madeira and 21 of February in central mainland Spain, when it clearly overestimated dust concentrations (see orange circles in Figure 6A1-6A4). A few days later, 23-28 February 2021, another dust event impacted across central to northern and eastern Europe due to eastward shift the L-to-H dipole and the resulting northward dust transport across the central Mediterranean (Meinander et al., 2023; Peshev et al., 2023).

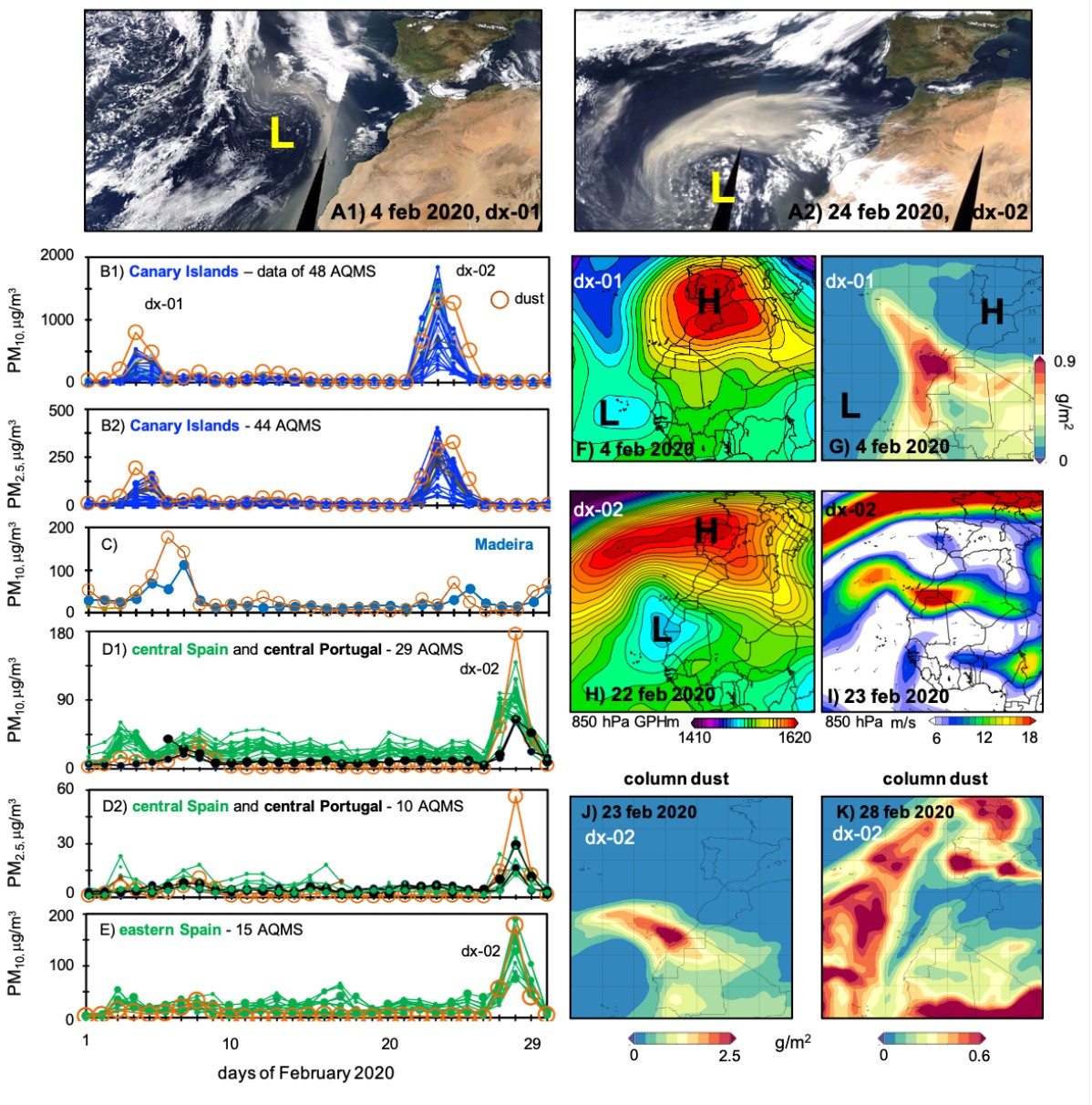

**Figure 4. Events dx-01 and dx-02.** Satellite view (NOAA-20 VIIRS) of the dust plume during the first day of dx-01 (A1) and dx-02 (A2) events. Time series of (24h average) $PM_{10}$ and $PM_{2.5}$ data recorded in AQMS of the Canary Islands (B1-B2), Madeira island (C), central Portugal and Spain (D1-D2) and eastern Spain (E), which includes dust and $dust_{2.5}$ concentrations ($\mu g/m^3$) obtained with MERRA-2 model (orange circle) in each region (27-29ºN, 15-17.5ºW domain for the Canary Islands, 32-34 ºN, 16-18ºW) for Madeira island, 39-41ºN, 9.2-4.3ºW for central Portugal and Spain and 37-41.5ºN, 6.8ºW-1.2ºE for eastern Spain). The geopotential height (GPH) of the 850hPa is shown for the first day of dx-01 (E) and dx-2 (G) events. Wind vector at 850hPa for the 23 Feb 2020 dx-02 (E). MERRA-2 column dust load for the 04 Feb 2020 (F), 23 Feb 2020 (I) and 28 Feb 2020 (J).

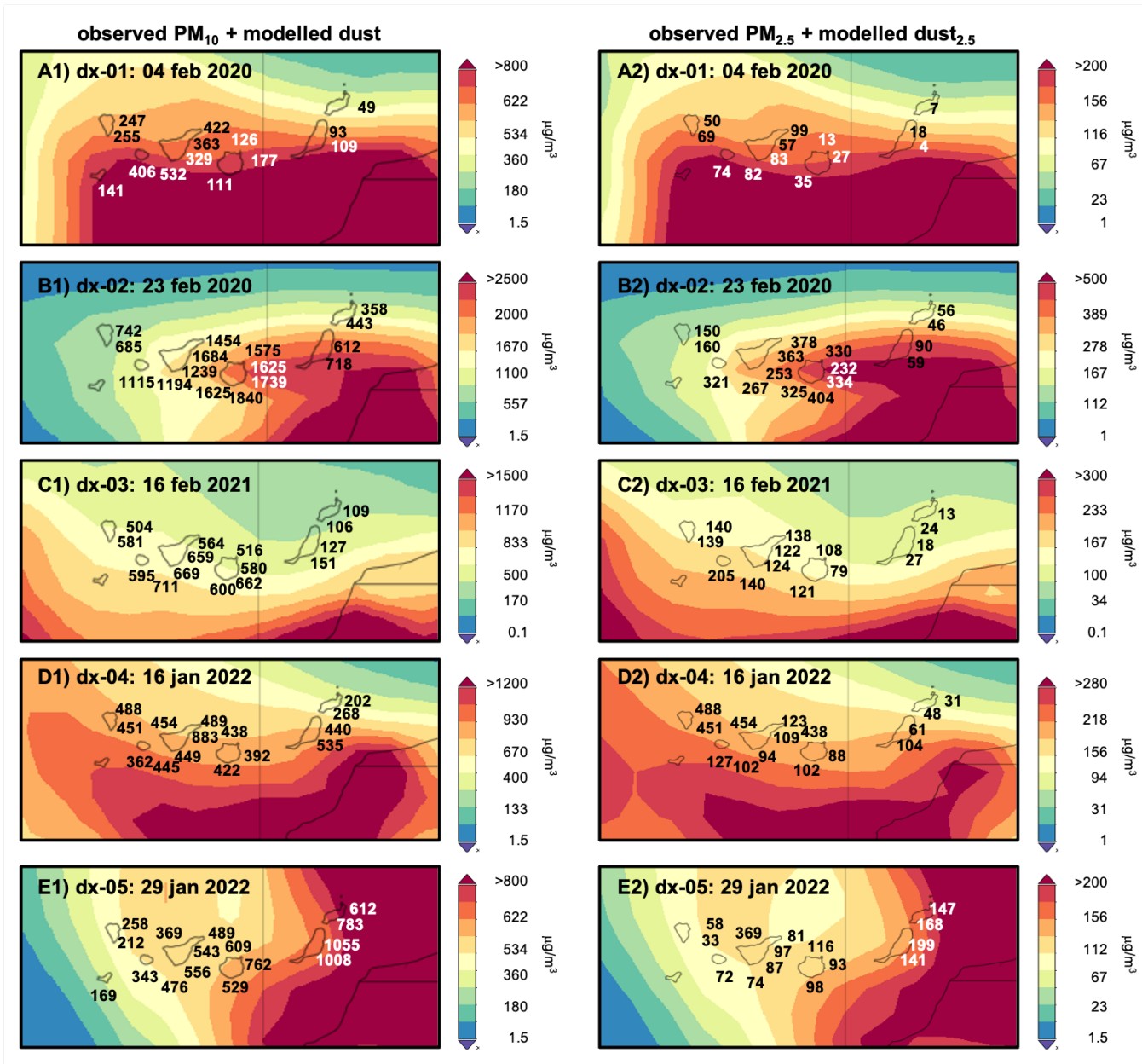

**Figure 5. Surface dust and dust$_{2.5}$ concentrations of MERRA-2 reanalysis (26.5-30.0ºN, 19.3-12.0ºW) and observed (24h average) PM$_{10}$ and PM$_{2.5}$ measured in AQMS during specific days of dx-01, dx-02, dx-03, dx-04 and dx-05 events in the Canary Islands.**

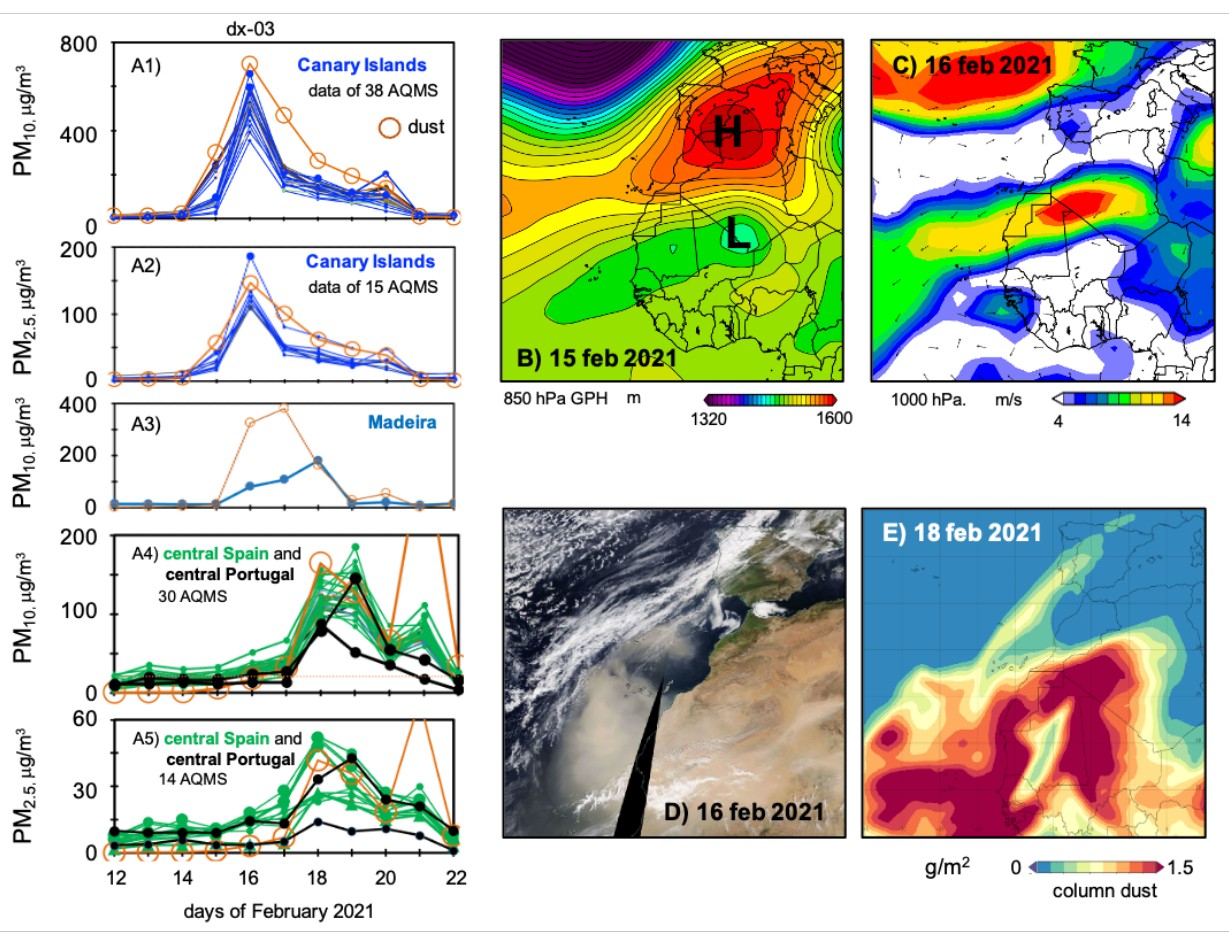

**Figure 6. Event dx-03.** Time series (A1-A4): (1) of (24h average) $PM_{10}$ and $PM_{2.5}$ in AQMS of the Canary Islands, Madeira, central mainland Portugal and central mainland Spain, and (2) of dust and dust$_{2.5}$ concentrations ($\mu g/m^3$) obtained with MERRA-2 in the Canary Islands (27-29ºN, 15-17.5ºW), Madeira island (32-34 ºN, 16-18ºW), central mainland Portugal and Spain (39-41ºN, 9.2-4.3ºW). The geopotential height (GPH) of the 850hPa level (B, 15 feb 2021), wind at 1000hPa (C, 16 feb 2021), the satellite image (NOAA-20 VIIRS)(D, 16 feb 2021) and the column dust load (E, 18 feb 2021) are included.

Another two extreme dust events occurred in January 2022 (Fig.7). The dx-04 event impacted on the Canary Islands during 14-17 January 2022 (Fig.7A1), resulting in (24h average) $PM_{10}$ and $PM_{2.5}$ concentrations within the range 275-883[x] $\mu g/m^3$ (Fig.7A1; Fig.5D1)([x]Tenerife, Casa Cuna AQMS) and 60-136[x] $\mu g/m^3$ (Fig.7A2; Fig.5D2) ([x]Tenerife, Caletillas AQMS) and 1-hour average $PM_{10}$ and $PM_{2.5}$ concentrations reaching values within 1200-2170 $\mu g/m^3$ and 240-550 $\mu g/m^3$ across the Canaries (Fig.S1D1-D2), respectively. A few days later, the dx-05 occurred (29 Jan - 1 Feb 2022; Fig.7A2-7A3), impacting again the Canary Islands, resulting in (24h average) $PM_{10}$ and $PM_{2.5}$ concentrations within the range 314-1055[x] $\mu g/m^3$ (Fig.7B1; Fig.5E1)([x]Fuerteventura, El Charco AQMS) and 70-199[x] $\mu g/m^3$ (Fig.7B2; Fig.5E2) ([x]El Charco AQMS) and 1-hour

average PM$_{10}$ and PM$_{2.5}$ concentrations reaching values within 1000-2520 μg/m$^3$ and 400-545 μg/m$^3$ in the eastern islands (Fig.S1E1-E2), respectively. In Madeira, PM$_{10}$ concentrations ranged within 80-225 μg/m$^3$ during dx-04 and dx-05 (Fig.7B3). In both cases a massive dust plume was transported northward, approaching to Spain and Portugal (Fig.7D1-7D2). As in the previous cases, these duxt episodes were caused by a L-to-H dipole meteorology, with an anticyclonic core over Europe expanding to North Africa and a cyclone south of the Canary Islands (Fig.7C1-7C2). In the two events, MERRA-2 clearly overestimated dust concentrations (Fig.7B1-7B2), suggesting that the model may be transporting dust at too low altitude (O'Sullivan et al., 2020).

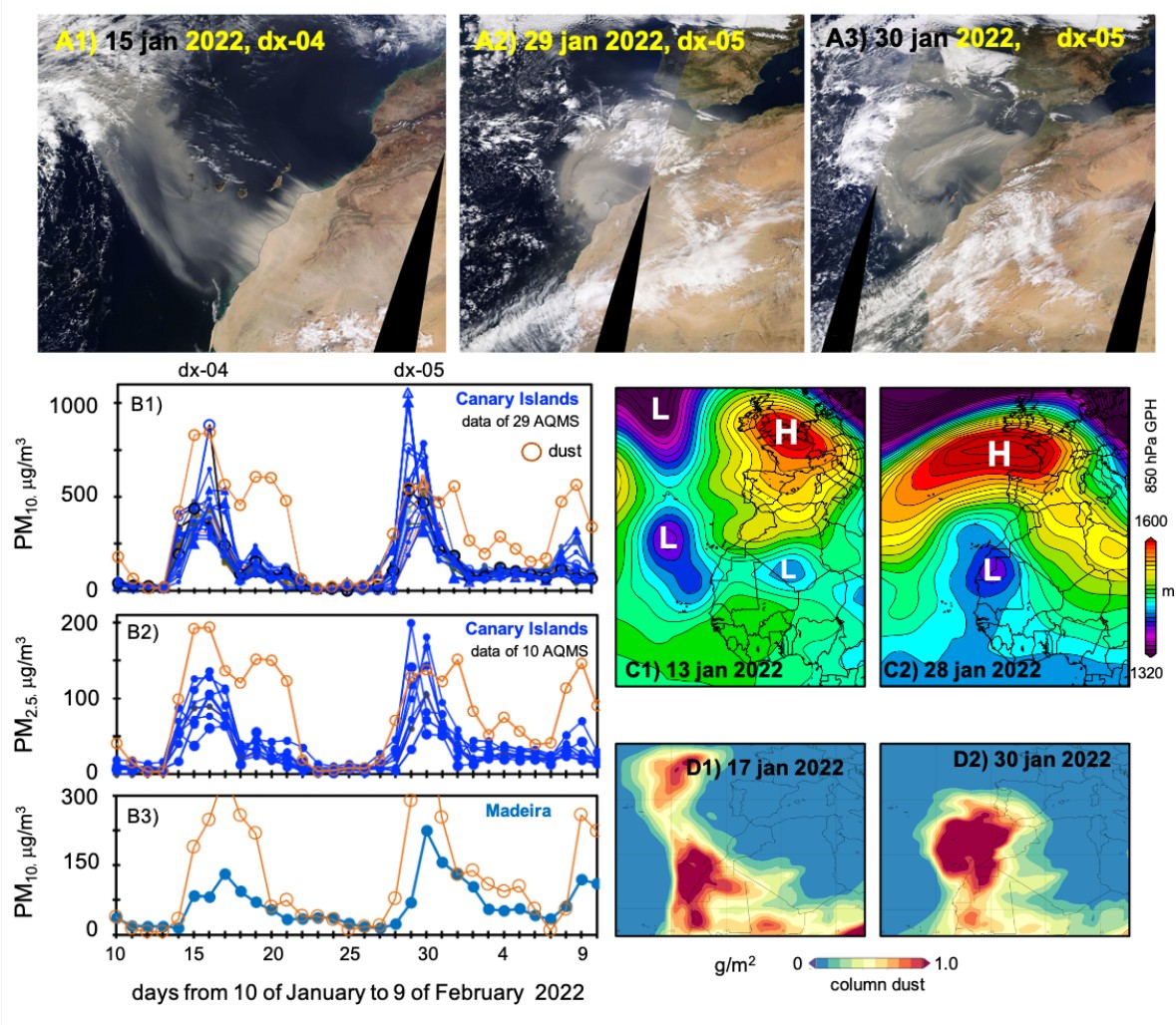

**Figure 7. Events dx-04 and dx-05. Satellite view (NOAA-20 VIIRS) of the dust plumes (A1-A3). Time series of (24h average) PM$_{10}$ and PM$_{2.5}$ in AQMS of the Canary Islands (B1-B2) and Madeira island (B3) and MERRA-2 surface dust and dust$_{2.5}$ concentrations in the Canary Islands (27-29ºN, 15-17.5ºW) and Madeira (32-34 ºN, 16-18ºW). The geopotential height (GPH) of the 850hPa level (C1-C2) and column dust load (D1-D2) are included.**

Finally, the sixth duxt event (dx-06: 15-20 March 2022) first impacted mainland Spain and Portugal and subsequently the Canary Islands (Fig.8). The event was also prompted by a meteorological L-to-H dipole, linked to the location of the cyclone *Celia* over Morocco and an anticyclonic core over the central Mediterranean (Fig. 8B). The resulting dusty jet (Fig. 8C-8E) expanded from southeast to northwest mainland Spain and Portugal on the 15 and 16 March 2022. Once in northern Spain, the

5 dust plume split in two branches, a branch travelled eastward across central Europe tracking the anticyclonic circulation at the north of the high-H (Qor-El-Aine et al., 2022), the other branch travelled southward over mainland Portugal to the Canary Islands tracking the cyclonic L circulation (Fig.8E-8F).

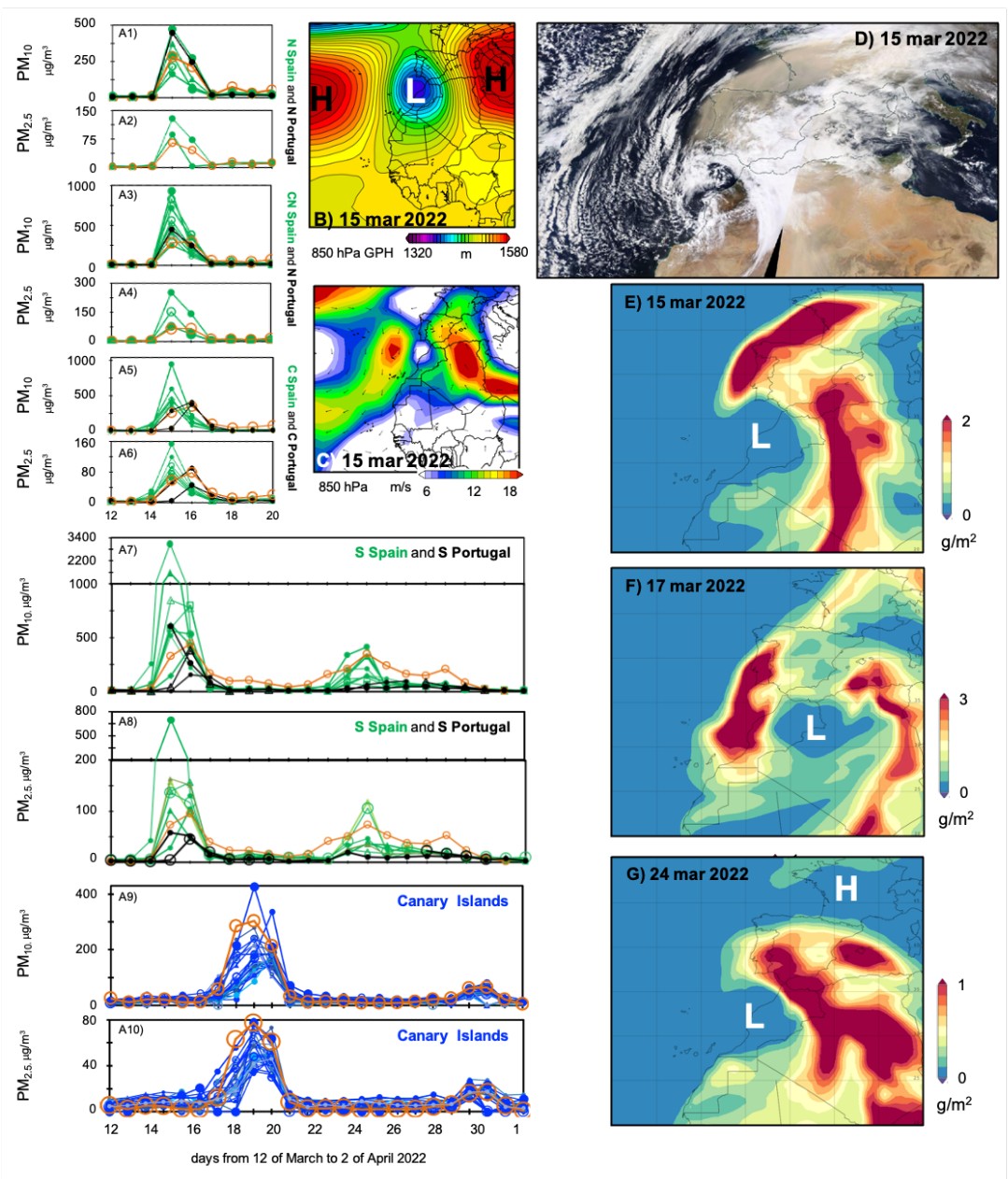

**Figure 8.** Event dx-06. Time series of (24h average) PM₁₀ and PM₂.₅ in AQMS of the Canary Islands (**blue time series**) and different regions of mainland Spain (**green time series**) and mainland Portugal (**black time series**): A1-A2) Northern (N) Spain (Cantabria and Galicia regions) and Portugal (Norte). A3-A4) Central-Northern (CN) Spain (Castilla y León region) and Portugal (Norte); A5-A6) Central (C) Spain (Madrid + Extremadura + Castilla La Mancha) and Portugal (Centro + north Alentejo); A7-A8) Southern (S) Spain (Andalucía) and Portugal (south Alentejo and Algarve); A9-A10) Canary Islands. The plots also include surface dust concentrations in these regions obtained with MERRA-2 reanalysis (A1-A10, **orange time series**). The geopotential height (GPH) (B) and wind (C) in the 850hPa level, the satellite NOAA-20 VIIRS image and column dust load (E-G) is also included.

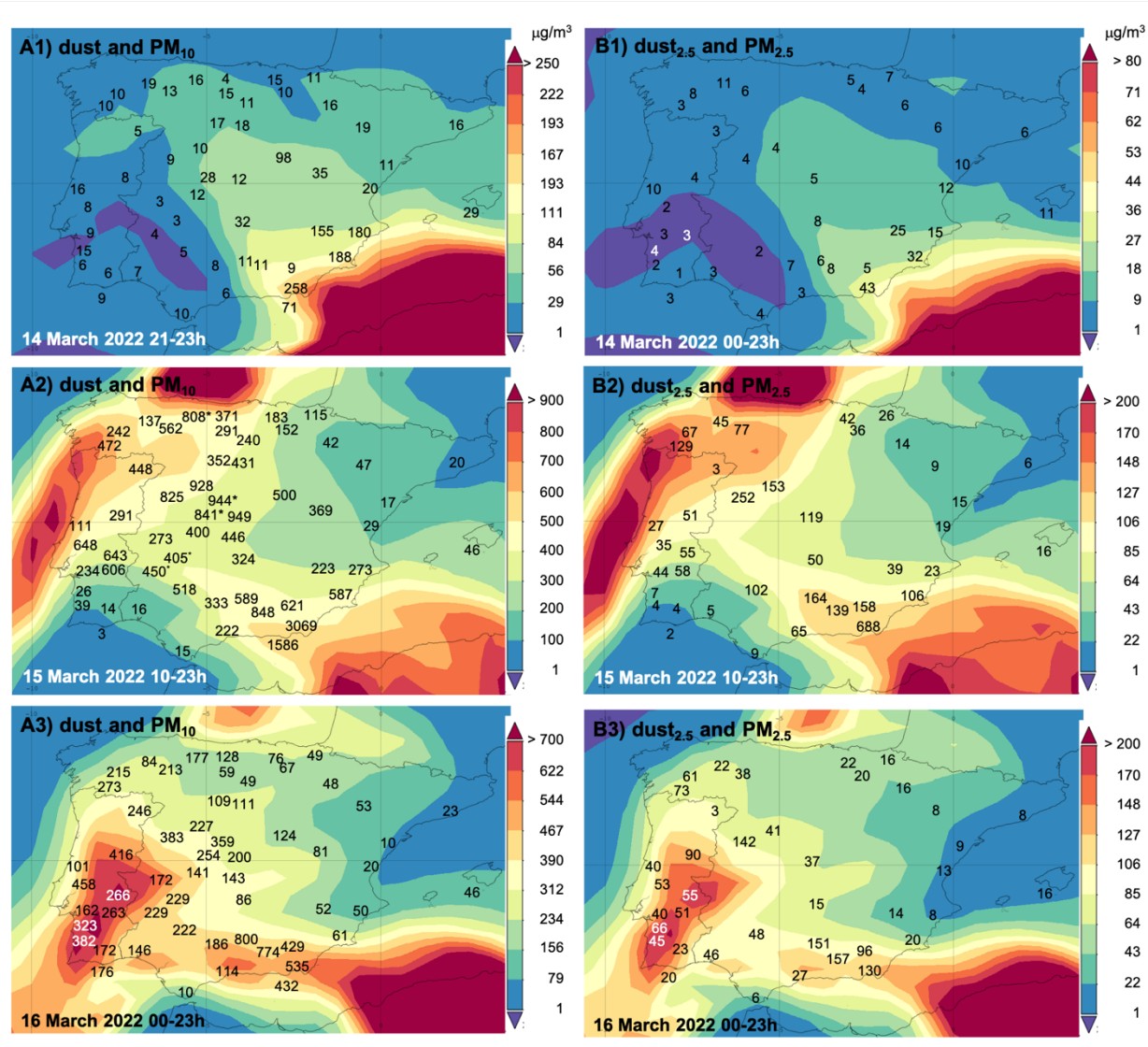

**Figure 9. Surface concentrations of dust (A1-A3) and dust particles smaller than 2.5 microns (dust₂.₅) (B1-B3) from 14 to 16 March 2022 in mainland Spain according to MERRA-2 reanalysis. Daily (24h) average concentrations of PM₁₀ (A1-A2) and PM₂.₅ (B1-B2) measured in AQMS are shown with black numbers.**

In Iberia, $PM_x$ concentrations experienced a sharp increase, from their regular background levels (10-30 μg/m³) to 24h average $PM_{10}$ and $PM_{2.5}$ values within the range (i) 500-3070 μg/m³ and 100-700 μg/m³ in southern regions of Spain and Portugal (Murcia, Andalucía, Algarve and Alentejo) (Fig.8A7, 8A8, and 9), (ii) 200-1000 μg/m³ and 60-160 μg/m³ in central parts of Spain and Portugal (Castilla La Mancha, Madrid, Extremadura, Vale do Tejo and Lisboa) (Fig. 8A5, 8A6 and 9), (iii) 200-1000 μg/m³ and 60-260 μg/m³ in central northern Spain (Centro and Castilla y León) (Fig. 8A3, 8A4 and 9), and (iv) 150-500 μg/m³ and 75-130 μg/m³ in northern Portugal and Spain (Norte, Cantabria and Galicia) (Fig. 8A1, 8A2 and 9) during the 15-16 March 2022, respectively. In Lisboa, these extremely high $PM_{10}$ and $PM_{2.5}$ concentrations where even registered in the indoors environment (Gomes et al., 2022). After several days traveling across thousands of kilometres, the dust plume impacted in the Canary Islands during 17-20 March 2022 (Fig.8F), resulting in (24h average) $PM_{10}$ and $PM_{2.5}$ values within the range 150-430 μg/m³ (Fig.8A9) and 30-80 μg/m³ (Fig.8A10). A regular to intense (no extreme) dust event impacted southern Spain 24-25 March 2022 ($PM_{10}$ and $PM_{2.5}$: 50-420 μg/m³ and 30-120 μg/m³; respectively) (Fig.8A7-8A8) and the Canary Islands 24-25 March 2022 ($PM_{10}$ and $PM_{2.5}$: 40-80 and 20-35 μg/m³, respectively).

This historic dx-06 event (Fig.1D-1E) started the evening of 14 March 2022 (>21h), with a dust inflow in south-eastern Spain that led to 24h average $PM_{10}$ and $PM_{2.5}$ concentrations within the range 70-260 μg/m³ and 25-43 μg/m³ (Fig.9A1-9B1). On 15 March 2022, the massive dust plume moved from south-eastern Spain, where it led to 24h average $PM_{10}$ values within the range 580-3070 μg/m³, toward the west and northwest of Iberia, resulting in (24h average) $PM_{10}$ concentrations within the ranges 825-950 μg/m³ in central Spain, 600-650 μg/m³ in central Portugal and 450 440-810 μg/m³ in northern Portugal and northwest Spain (Fig.9A2). Concentrations of $PM_{2.5}$ (24h average) were within the range 139-690 μg/m³ in south-eastern Spain, 25-60 μg/m³ 40-70 central Portugal, 100-260 μg/m³ in central Spain and 50-130 μg/m³ in central-north Portugal and north-western Spain (Fig.9B2). These extremely high $PM_x$ concentrations were also recorded indoor (Gomes et al., 2022). During the 16 March 2022, high $PM_{10}$ and $PM_{2.5}$ values were still recorded (Fig.9A3-9B3), with the highest $PM_{10}$ concentrations linked to the still ongoing dust inflow by south-eastern Spain (800 μg/m³) and the southward transport of dust in southern Portugal (300-330 μg/m³). Because of the massive dust load the solar energy production in Spain dropped in a 50% (Micheli et al., 2024). Details on this event, as the maximum hourly $PM_{10}$ and $PM_{2.5}$ concentrations (Fig.S2) and the names of the AQMS, are provided in the Supplement.

### 3.3 Record breaking events

The analysis of the 2000-2022 time series of (24h average) $PM_{10}$ (Fig.10) and $PM_{2.5}$ (Fig.11) data, evidence that the duxt events we report here are record beating episodes in mainland Spain, continental Portugal and the Canary Islands. The massive

dusty airmass that blackened the Iberian Peninsula during the 15 and 16 of March 2022 (dx-06; Fig.10 and 11) resulted in the highest $PM_{10}$ and $PM_{2.5}$ concentrations ever recorded in the regional scale across northern Spain (Cantabria region; Fig.10A), central - northern Spain (Castilla y León region; Fig. 10C & 11A), central Spain (Castilla La Mancha, Extremadura and Madrid region; Fig.10D & 10A), southern Spain (Andalucía region; Fig. 10E, 10F & 11C) and continental Portugal (Fig.10G). In

5    central Spain (Castilla La Macha + Extremadura + Madrid), regular Saharan dust events typically induce (24h average) $PM_{10}$ concentrations within the range 40-140 $\mu g/m^3$ (highlighted with black arrows in Fig.10D) (Pey et al., 2013; Rodríguez et al., 2001), anomalous intense dust events as that occurred 22 February 2016 (Sorribas et al., 2017) resulted in $PM_{10}$ concentrations 200-380 $\mu g/m^3$, whereas during the dx-06 event 33 AQMS of this region recorded (24h) $PM_{10}$ concentration within 300-949[x] $\mu g/m^3$ ([x]Villa del Prado AQMS in Madrid region)(white arrow in Fig. 10D). After analysing the 2001-2011 time series, Pey et

10    al.(2013) concluded that in the Western Mediterranean, Saharan dust events inducing $PM_{10}$ >100 $\mu g/m^3$ (24h average) are actually rare. The impact of the dx-06 event is not observed in Cataluña since it did not reach Northeast Spain (Fig.10B).

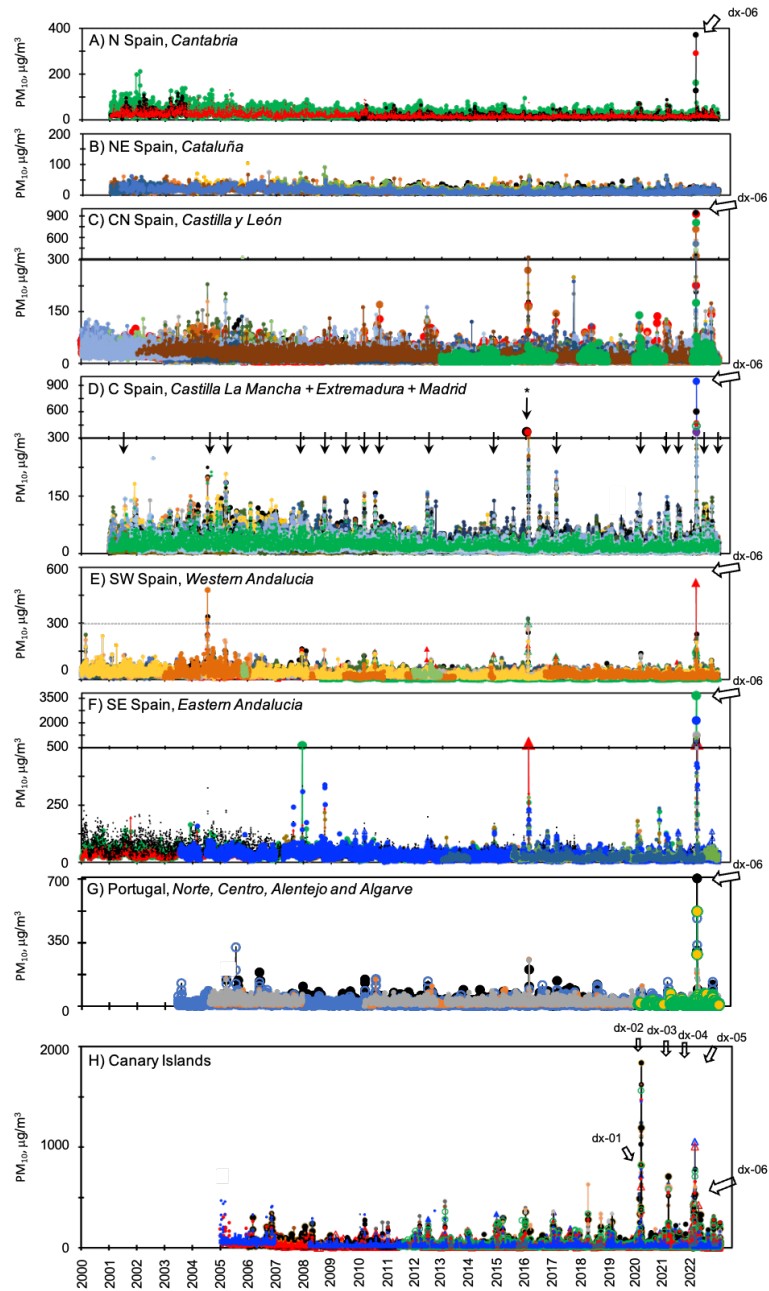

**Figure 10. Time series (2000-2022) of (24h average) PM$_{10}$ concentrations in 123 AQMS distributed across Portugal (5 AQMS), northern (N; 3 AQMS), North-East (NE; 8 AQMS), Central-North (CN; 30 AQMS), Central (C; 35 AQMS), Southwest (SW; 16 AQMS) and Southeast (SE; 11 AQMS) mainland Spain and in the Canary Islands (15 AQMS). Black arrows indicate regular dust events. White arrows indicate the duxt events. The asterisk indicates the intense event occurred 22 Feb 2016.**

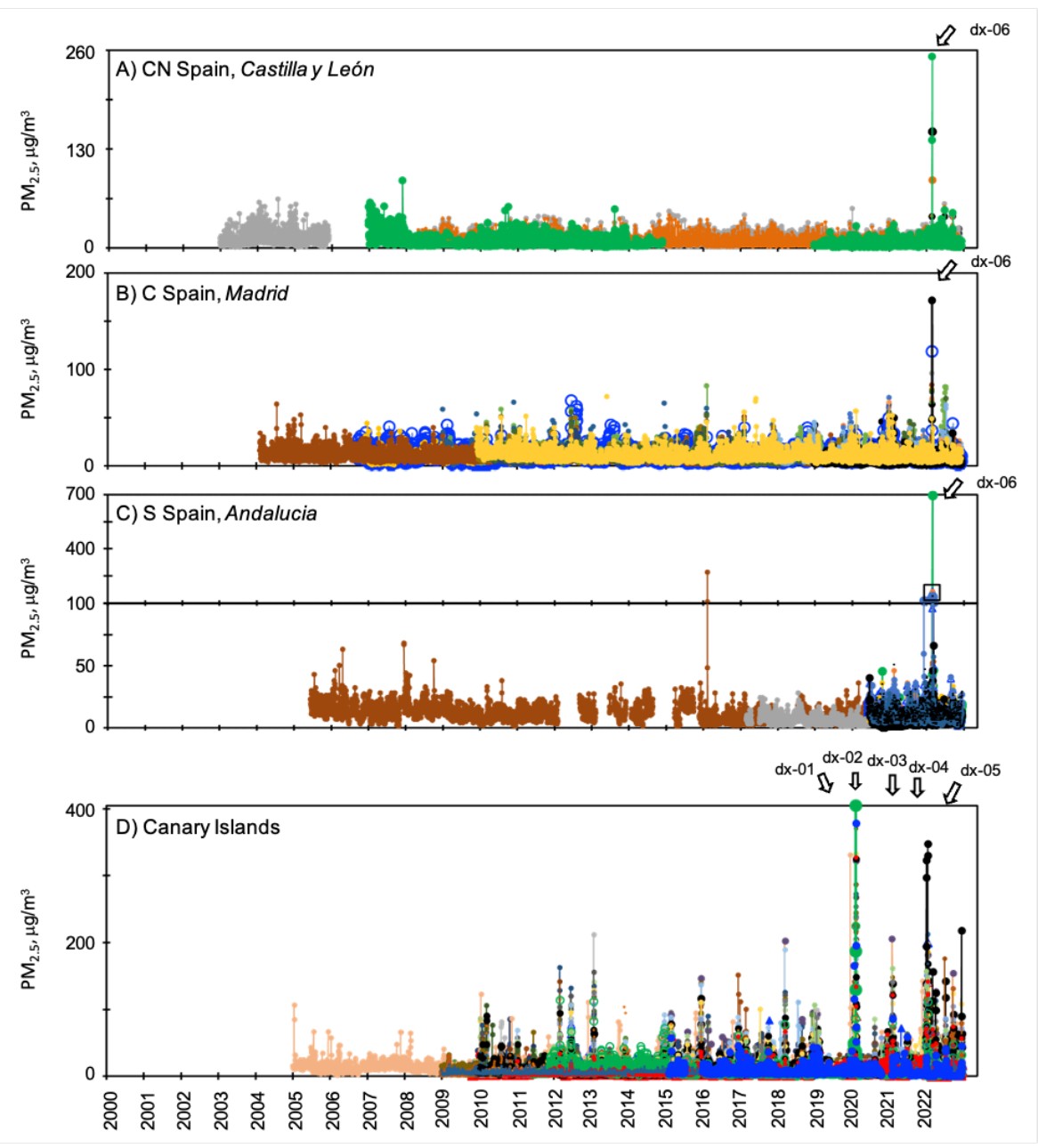

**Figure 11. Time series (2000-2022) of (24h average) PM2.5 concentrations of in a total of 74 AQMS distributed across the Central-North (CN; 4 AQMS), Central (C; 16 AQMS) and Southeast (S; 10 AQMS) mainland Spain and in the Canary Islands (44 stations). Black arrows indicate regular dust events. White arrows indicate the duxt events.**

In the Canary Islands, the most intense Saharan dust events ever recorded occurred in the period 2020-2022 linked to the duxt events described in this study. Since 2005 to 2020, intense Saharan dust events have regularly been associated with (24h average) $PM_{10}$ and $PM_{2.5}$ values with 200-400 µg/m³ (Fig.10H) and 100-200 µg/m³ (Fig.11D), respectively. This $PM_{10}$ range is similar to (1) that of total suspended particles during Saharan dust events of the period 1998-2003, based on AQMS data not yet normalised to the European standards (Alonso-Pérez et al., 2007, 2012; Viana et al., 2002), and to (2) that of total dust at Izaña Observatory (Tenerife) during the 1987-2014 Saharan dust events (Rodríguez et al., 2015). In contrast, in the period 2020-2022, the duxt events described here led to $PM_{10}$ and $PM_{2.5}$ concentrations within the ranges (24h average) 600-1840 µg/m³ (Fig.10G) and 200-404 µg/m³ (Fig.11D), respectively. The three most intense dust events ever recorded in the Canary Islands, exceeding the threshold of 600 µg/m³ of $PM_{10}$ as 24h average, in descending order of magnitude, are:

1. dx-02 event: the 23 and 24 February 2020 the (24h average) $PM_{10}$ averaged in all AQMS were 531 and 930 µg/m³ (averages of 34 AQMS distributed in the 7 Canary Islands), respectively. The 22, 23 and 24 February 2020, a total of 6, 25 and 12 AQMS recorded a (24h average) $PM_{10}$ concentration between 600 and 1840[x] µg/m³ ([x]Gran Canaria, Playa del Inglés AQMS). Previous to this event, the (24h average) $PM_{10}$ concentrations had only exceeded 600 µg/m³ in just one AQMS (618 µg/m³ at Las Galanas during the dust event 28 March 2018; Fig.10G).

2. dx-05 event: 29 and 30 January 2022 the (24h average) $PM_{10}$ averaged in all the AQMS was 463 and 501 µg/m³ (averages of 44 AQMS distributed in the 7 islands). The 29 and 30 of Jan 2022, a total of 9 and 10 AQMS recorded a (24h average) $PM_{10}$ concentration between 600 and 1055[x] µg/m³ ([x]Fuerteventura, El Charco AQMS).

3. dx-03 event: the 16 February 2021 the (24h average) $PM_{10}$ averaged in all AQMS was 463 µg/m³ (average of 36 AQMS distributed in the 7 islands). The 16 February 2021, 9 AQMS recorded a (24h average) $PM_{10}$ concentration between 600 and 711[x] µg/m³ ([x]Las Galletas AQMS, Tenerife).

In mainland Spain and continental Portugal, the dx-06 is the most intense dust event ever recorded. In Spain total of 20 AQMS, distributed across south-eastern, central to central-northern Spain, registered (24h average) $PM_{10}$ concentrations within the range 600 to 3070[x] µg/m³ ([x]Mediterraneo AQMS in Almería province). In Portugal, 4 AQMS located in the central regions registered (24h average) $PM_{10}$ concentrations within the range 600 to 648[x] µg/m³ ([x]Chamusca AQMS in Lisboa - Vale do Tejo). The 2 decades time series of $PM_x$ also offer other interesting data. In many of regions of Spain there is a clear 2000-2020 decreasing trend of $PM_{10}$ concentrations linked to the reduction of emissions following air quality policies (Fig.10A, 10C, 10E) (Li et al., 2018; Querol et al., 2014), suggesting that desert dust may have an increasing relative contribution to $PM_{10}$ concentrations as also pointed by recent projections (Gomez et al., 2023).

## 3.4 Meteorological anomalies linked to duxt events

In winter, North African dust is regularly transported southward to tropical latitudes (Merdji et al., 2023). In this season, dust transport northward, to the subtropical North Atlantic and southern Europe, occurs during rather short periods, under specific meteorological scenarios described in previous studies (Flaounas et al., 2015; Fluck and Raveh-Rubin, 2023a; Rodríguez et al., 2001). Winter extreme Saharan dust events have been observed in the southern Sahara – Sahel region, from Mauritania to Niger along the Sahel, associated with $PM_{10}$ and $PM_{2.5}$ concentrations within the ranges $800 – 5000$ μg/m$^3$ and $600 – 1300$ μg/m$^3$ (Fluck and Raveh-Rubin, 2023b; Marticorena et al., 2010), respectively, induced by H-to-L dipoles, i.e. an anticyclonic high pressure H core over Atlantic and western North Africa and a cut of low L over the Mediterranean, a meteorological configuration that results in strong southern winds (Fluck and Raveh-Rubin, 2023b).

The 2020-2022 six duxt events we report here were induced by the L-to-H meteorological dipoles formed by cyclones located at the southwest of a blocking anticyclone over western Europe. Fig. 12 (A1 to F1) show the anomaly of the 500 hPa geopotential height during the onset of the six duxt events (Fig.12 A2 to F2). All duxt events occurred during northern-hemisphere meteorological anomalies that resemble the anomalies of the atmospheric circulation that have been linked, in previous studies, to global warming (Fig.12): (i) subtropical anticyclones expanded and shifted to higher latitudes (Cherchi et al., 2018; Cresswell-Clay et al., 2022), (ii) anomalous low pressures expanding northward beyond the tropical belt, resembling the tropical expansion (Seidel et al., 2008; Yang et al., 2020, 2023) (e.g. dx-01, dx-02, dx-03 and dx-04) and (iii) mid-latitudes amplified Rossby waves due to the concatenation of cut-off lows cyclones and anticyclones pointing to a weakening of the polar vortex (e.g. dx-03, dx-05, dx-06 and 23-Mar-2022) (Mann et al., 2017; Screen and Simmonds, 2013). The duxt events observed at the Eastern Mediterranean in March 2018 (Solomos et al., 2018) and in March 2020 (Mifka et al., 2023), at North Africa in June 2020 (Bi et al., 2023; Francis et al., 2020), at Uzbekistan in November 2021 (Xi et al., 2023) and in China in March 2021 (Gui et al., 2022; Liu et al., 2023), occurred in this context of cyclones, blocking anticyclones and dipoles linked to mid-latitudes amplified Rossby waves.

The winter blocking anticyclone that we observed over western Europe/Western Mediterranean during the duxt events (Fig.12A1-G1) fits with the picture of the industrial-era eastward expansion and shift of the North Atlantic anticyclone starting in the 1850s and accelerating in the last few decades (Alonso-Pérez et al., 2011b; Cresswell-Clay et al., 2022), a trend that is expected to continue as the concentrations of greenhouses gases increase, according to the CMIP5 multi-model simulations (Cherchi et al., 2018); the low pressures at the southwest of the blocking anticyclone that we observe during the duxt events (Fig.12B1-12E1) are expected to follow as the tropic expands northward off North Africa in the forthcoming decades (e.g. see Fig 1F of Cherchi et al. 2018 for the 2075-2100 period).

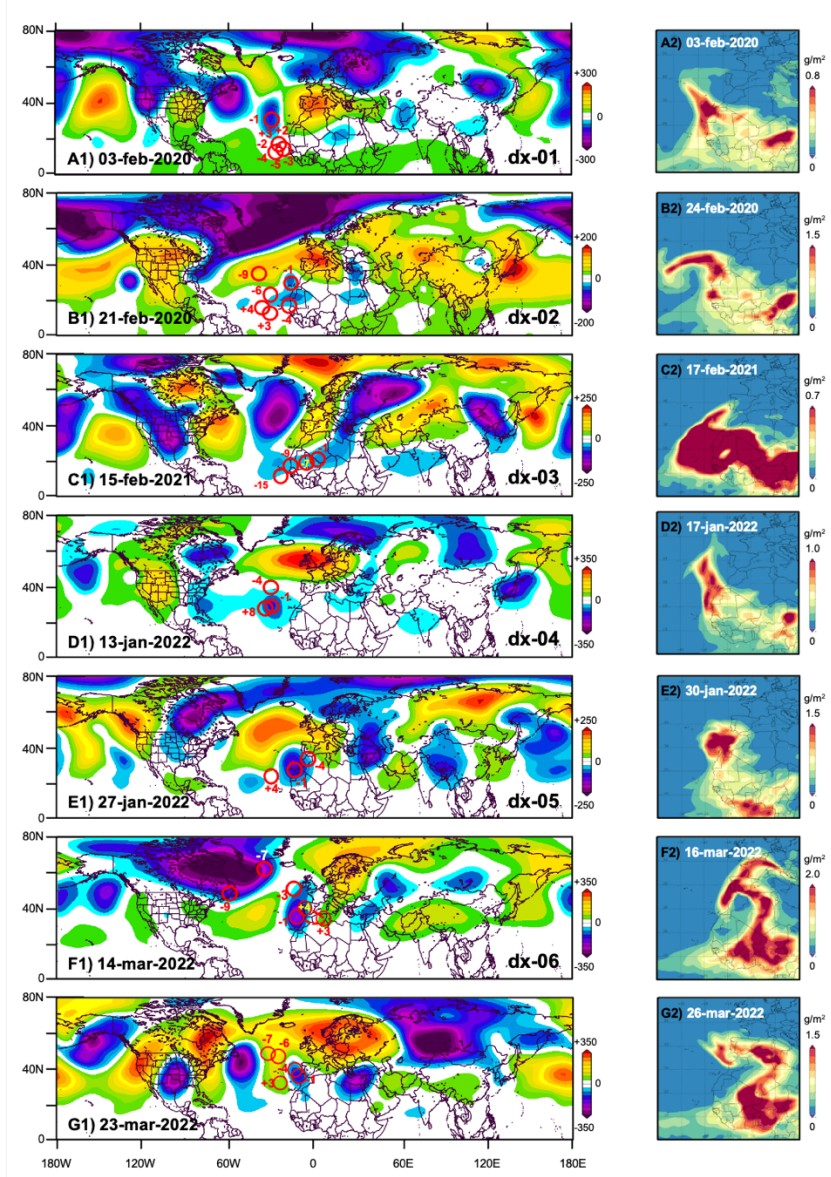

**Figure 12. Anomaly (A1-G1) of the 500 hPa geopotential height (GPH) (respect to the 1991-2020 climatology) and column dust in the onset of the dx-01 to dx-06 and 24-26 March 2022 events (A2-G2). Red circles (A1-G1) indicate the location of the cyclones days before (negative number) to days after (positive number) the first day of the duxt event.**

The anomalies linked to the duxt events are also evident in the trajectory of the cyclones that finally form the L-to-H dipoles leading to the extreme dust events. Red circles in Fig.12A1-12G1 indicate the location of the cyclones from days before (negative number) to days after (positive number) of the duxt event. Due to the anticyclonic blocking over western Europe, the mid-latitudes North Atlantic cyclones didn't follow their regular path across Europe or the Mediterranean. The cut-off lows forming the L-to-H dipoles reached this region (Canary Islands, Cape Verde or inner Sahara) by two main paths:

1. they deviated southward from the regular mid-latitudes westerly circulation in the North Atlantic as results of the blocking anticyclone over western Europe; subsequently these cut-off lows may stay during long periods (up to 12 days) in the subtropical and tropical north-east Atlantic (near the Canary Islands and Cape Verde). This is the case of the dipoles of events dx-02, dx-04, dx-05 and dx-06, and also of the intense dust event of 23 March 2022. The cyclones of the events dx-02, dx-04 and dx-05 stayed (blocked by the anticyclonic situation) near the Canary Islands and Cape Verde during 10, 12 and 8 days (Fig. 12B1, 12D1 and 12E1), respectively.

2. they deviated northward from the tropical belt. This is the case of the events dx-01 and dx-03, associated with cyclones which had moved from Cape Verde to the west of the Canary Islands (Fig.12A1, dx-01) and to the inner Sahara (Fig.12C1, dx-03), respectively, across a tropical band anomalously shifted to northward.

The observed sharp increase in dust transport to the western Euro-Mediterranean region in the 2020-2022 winters has been also been associated to the meteorology linked to dipoles and blocking anticyclones (Cuevas et al., 2023).

## 4 Summary and conclusions

In winter 2020, 2021 and 2022 a set of six extreme dust events expanded northward from NW Africa to the Atlantic and Europe, causing extremely high concentrations of $PM_{10}$ and $PM_{2.5}$ in the Governmental Air Quality Monitoring Stations of the Canary Islands, mainland Spain and continental Portugal, exceeding the upper operation limit of many $PM_{10}$ monitors. We developed the *duxt-r* methodology for assessing the consistency of the $PM_{10}$ and $PM_{2.5}$ data and re-construct the underestimated $PM_{10}$ concentrations. During these extreme dust events, the 1-hour average $PM_{10}$ and $PM_{2.5}$ concentrations were within the range 1000-6000 $\mu g/m^3$ and 400-1200 $\mu g/m^3$, whereas the 24-h average $PM_{10}$ and $PM_{2.5}$ data were within the range 500-3070 $\mu g/m^3$ and 200-690 $\mu g/m^3$, respectively. These extreme dust episodes were caused by the intense winds associated with the meteorological dipoles formed by a blocking anticyclone over western Europe and a cut-off low located at the southwest, near the Canary Islands, Cape Verde or into the Sahara. The analysis of the 2000-2022 time series of $PM_{10}$ and $PM_{2.5}$ shows that these events have no precedent. Record beating $PM_{10}$ and $PM_{2.5}$ (24h average) concentrations were measured in the Canary

Islands during the 22-24 February 2020, 1840 μg/m$^3$ and 404 μg/m$^3$, and in mainland Spain during 15-16 March 2022, 3069 μg/m$^3$ and 688 μg/m$^3$ (Almeria province) and 648 and 90 μg/m$^3$ in central Portugal, respectively. All duxt events occurred during northern-hemisphere meteorological anomalies associated with subtropical anticyclones shifted and expanded to higher latitudes, anomalous low pressures expanding beyond the tropical belt and a concatenation of cut-off lows and anticyclones

suggesting to a weakening of the polar vortex. Climate projections forecast the expansion of the North African drylands toward the northwest, increasing the risk of desertification of Spain and Portugal, with an associated increase in regional dust loads. The air quality monitoring networks need to adapt the strategy and operation range of the PM$_{10}$ and PM$_{2.5}$ monitoring programs to ensure accurate measurements during these extreme dust events due to the importance of having suitable data in the public data sets for health effects studies, modelling, etc…New studies have reported on recent record beating PM$_{10}$ and PM$_{2.5}$

episodes linked to dipoles induced extreme dust events from North Africa and Asia, in a paradoxical context of multidecadal decrease of dust emissions, a topic that will require further investigations.

**Code and Data availability**

The duxt-r "PMx evaluation and reconstruction method based on ratios during extreme dust events" methodology descried in

this study is registered in Blockchain – SigneBlock. The data used in the manuscript are available in public data bases, including the PM$_{10}$ data reconstructed as result of this study. Data are also available at DIGITAL CSIC (http://hdl.handle.net/10261/364553) and by request to the first author at sergio.rodriguez@csic.es.

**Author contributions**

SR and JLD performed the conceptualisation, investigation, data collection, treatment and formal analysis. SR wrote the

original version of the manuscript, which was subsequently revised and edited by SR and JLD.

**Competing interests**

One of the authors [SR] is editor in the journal Atmospheric Chemistry and Physics.

**Acknowledgments**

We thanks to the 17 Autonomous Regions of Spain and the Ministry for the Ecological Transition and the Demographic

Challenge for the kind supply to the data of the air quality monitoring stations. We also thanks to the Agência Portuguesa do Ambiente for supplying the air quality data of Portugal. We acknowledgement the access to modelling reanalysis data by the

portal web sites of NASA - Giovanni and NOAA – Physical Science Laboratory. Satellite images were provided by the NASA World View website. We also thanks to Jon Vilches (air quality management in the Government of the Canary Islands) for the interesting talks on the $PM_x$ monitoring devices.

**Financial support**

5   This study is part of the project AERO-EXTREME (PID2021-125669NB-I00) funded by the State Research Agency/Agencia Estatal de Investigación of Spain and the European Regional Development Funds.

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
