# Peer review of "Extreme Saharan-dust events expand northward over the Atlantic and Europe prompting record-breaking PM10 and PM2.5 episodes"

_EGUsphere, 2023_

## Author Response (AR1)

**Author's response**

**Reviewer 1**

**R1-C1**

The paper presents an exhaustive analysis of extreme dust episodes that occurred in the last 4 years in wester europe.The episodes are relevant and the analysis is quite complete. However there are no innovative methods developed or new challenges to handle. I am favour of publication after some minor and major changes to be addressed. See below my comments.

**RE:** thank you for this comment. This study includes an innovative method for the analysis of the consistency of the PM10 and PM2.5 data and for the data reconstruction if needed. In our modest opinion this is important in order to have suitable PM10 and PM2.5 data in public datasets used, among others, for model constrain, long term modelling for understating long term dust variably, and for health effects studies (since the risk of mortality is determined for the increase in 10 ug/m3 of PM10).

Abstract (page 1)

**R1-C2**

Line 8: "occurred on 3-5 February..."

**RE:** thanks for this correction, which has been introduced in the revised version of the manuscript.

**R1-C3**

Line 13: a novel method is mentioned but nothing is said about it. The authors should indicate the basis of the method.

**RE**: not details were provided in the abstract in order to keep it as short as possible.

**R1-C4**

Line 25: Why only monitoring stations from Spain territory was analysed? It would be interesting to extend this analysis to other countries/regions that were also affected, e.g. Portugal

**RE:** this is a very very interesting suggestion. When we did the analysis of the data of Spain, we also thought in including PM10 and PM2.5 data of Portugal, however they were not yet available in the public data bases; now they are (from the Agência Portuguesa do Ambiente), we have downloaded the 1h resolution data and we have assessed and reconstrued some data. Following the suggestion of the reviewer data of Portugal are included in the revised version of the manuscript.

Methodology

**R1-C5**

Page 4, Section 2.2: more details should be given on the use of these satellite data (which paremeters; which time and spatial coverage...)

**RE:** we only used satellite images (not aerosol optical depth etc...), those shown in Fig. 4A1-4A2, 6D, 7A1-A3 and 8D. This is now described in the revised version of the manuscript.

Results and discussion

**R1-C6**

Page 7, Line 30: The asssumptions behind this method have to be identified and discussed, like, the authors consider that the low variability of the ratio PM2.5/PM10 obtained with the stations with available data allows to consider and apply an average value to the reconstruction of the time series in stations where data is not available

**RE**: Thanks for this comment. In this new method we did not assume any PM2.5/PM10 value. The evolution of PM2.5/PM10 ratio is calculated by interpolation, between the last valid PM10 value (just before saturation) and the first valid PM10 value (just after saturation), measured in each air quality station during each day and hour. We used an interpolation based on a linear evolution of the PM2.5/PM10 ratio with time, as shown in Fig. 3B (green line). The validation shown in Fig3E1-3E4 shows that this is a realistic approach. This (linear interpolation) is now described in the in the revised version of the manuscript.

**R1-C7**

Page 10, Line 10: only here it is explained how the comparison between reconstruction and measurements were done (Figure 3). This should be discuss before presenting Figure 3

**RE**: thanks for this comment. Following this suggestion, in the revised version of the manuscript, this is described before presenting the results, in the general description of the method.

**R1-C8**

Page 10, line 16: see "loos" instead of "loose"

**RE:** thanks for this correction, which is now included in the revised version of the manuscript.

**R1-C9**

Page 11, Line 1-4: please be consistent writing the name of the months

**RE:** thanks. This has been corrected in the revised version of the manuscript; the short names (Feb instead is used only in figures, their caption and between brackets for the shake of brevity)

**R1-C10**

Page 11, Line 8: The title of this section (3.2) is not clear, do not give any information about the content

**RE:** thanks for the comment. The title of the section was modified in order to described that it is an analysis of the events, rewritten as 3.2 Analysis of the extreme dust events

**R1-C11**

Page 11, Line 9-12: There is a repetition...please avoid this along the manuscript

**RE:** in order to avoid repetition, the first two lines were removed in the revised version of the manuscript.

**R1-C12**

Page 11, section 3.2: This section is characterized by an exhaustive descrition of each episode. It would be interesting to have some summary with the main characteristics of each episode and its comparison

**RE:** thanks for this comment. It is interesting. This summary is included in section 4.

**R1-C13**
Page 15, Figure 6: it should be mentioned that figures B, C, D and E are related to different days
**RE:** thanks for the suggestion, which has been included in the figures caption of the revised version of the manuscript.

**R1-C14**
Page 23, Lines 10-20: there is, again, repetition on the information of the episode values!
**RE:** in the revised version of the manuscript, a short reference to the dates which each dx event was included, in order to facilitated the traceability of the events by the readers. Dates were removed from other parts of the section (in order to avoid repetitions).

**R1-C15**
Page 30: It should be interesting to finish this section with same statistical quantification on the long time-series analysis (trend analysis)
**RE:** page 30 is part of the reference list, I guess that the reviewer means the section 3.4. This is included in section 4.

Thanks for the constructive and useful comments.

**Reviewer 2**

**R2C1**
Rodríguez and López-Darias report on six recent extreme dust events observed across mainland Spain and the Canary Islands, which they claim represent an emerging new trend of extreme Saharan dust events linked to global warming. The paper i) reports PM10 and PM2.5 concentration measurements from air quality monitoring stations (including a method for correcting saturated values during the extreme events), ii) provides analyses of the synoptic weather situations during the extreme events, and iii) provides a climatological assessment of the recent events and associated meteorological anomalies. My expertise is related to part i) and the in situ measurements of dust concentrations. I find this part of the paper to be generally well done. I am more skeptical about the final parts of the paper and feel the authors may need to limit some of their conclusions here.

I think the claims of 'record breaking' and 'a new phenomenon driven by global warming' probably need to be toned down. Climatologically speaking, the periods considered are short. For mainland Spain, there has only been 1 extreme event (dx-06) so far. For the Canaries, only a handful of events. I think its fair to hypothesize that this is the emergence of a new phenomenon potentially related to global warming. However the number of observed events is simply too small to make very strong conclusions, which is the impression I currently get from the paper.

**RE:** thanks for these comments, which definitively help us to improve the manuscript. An important nuance, we are not stating that these extreme dust events are unequivocally linked to global warming, we simply say that these events occur under meteorological anomalies that < resemble > (= take after, looks like, appear) to the anomalies of the atmospheric circulation linked to the anthropogenic global warming (anomalies identified in previous studies based on climate projections, studies cited in the manuscript); please see these details in (1) page 24, line 5-11 (section 3.4, line 5: …resemble…), (2) page 24, line 27 (section 4: …resemble…), (3) page 1, line 29, (abstract: …resemble…). We tried to be especially careful with this.

Understanding how climate change is affecting dust emissions, transport paths and the intensity of dust events is extremely important nowadays. For this reason, we considered that is necessary to do a brief comparison of the meteorological scenarios in which the extreme dust events occurs with the changes in atmospheric circulation attributed (in other studies) to global warming (section 3.4). In the introduction (page 2, lines 22-31; page 3, lines 1-2) we included a description of the state of knowledge on the influence of man action on dust emissions, including (1) a 56% increase in dust load the industrial respect to the pre-industrial times, and (2) a decrease in the dust load since the 1980s attributed to the slowdown in atmospheric circulation linked to global warming (see cited studies in the manuscript). This second point (2) makes the recent (2020-2022) extreme dust events of high interest, since they represent an abrupt change in the recent dust trend. The introduction also describes how the dry desert conditions of the subtropics are expected to shift northward (from North Africa to Spain and Portugal) due to the northward shift of the subtropical anticyclones (Cresswell-Clay et al., 2022; Guiot and Cramer, 2016, see details and references in the introduction). Dust records in the Canary Islands started in the 1987 at Izaña Observatory (Rodriguez et al., 2012 https://doi.org/10.1016/j.aeolia.2012.07.004 and Rodriguez et al., 2015 https://doi.org/10.5194/acp-15-7471-2015) and in mid 1990s (based on total suspended particles) and 2005 (based on PM10 and PM2.5) in the Canary Islands air quality network (references in the manuscript), which covers 36 years, a period in which concatenated extreme dust events as those we report have not occurred. Moreover, two recent dust – climate projections forecast an increase in the dust load in the North Atlantic in the next decades (see Liu et al., 2024 https://doi.org/10.1038/s41612-023-00550-9 and Gomez et al., 2023 https://www.nature.com/articles/s43247-023-00688-7).

In our study we do not attempt to propose any forecast of future dust events. For that reason, to use the term  in the title may cause some confusion, so, we are considering removing this term from the title in the revised version of the manuscript.

These nuances we have included in the revised version of the manuscript. Thanks (again) for these comments, which definitively contribute to improve the text.

**R2C2**
The synoptic and climatological analyses are interesting but I feel the conclusions drawn from these investigations might be overstretched. I reiterate that I am not an expert in this type of analysis. Nevertheless, there have been quite a few recent papers on this topic that the authors have not cited: e.g., Flaounas et al., 2015 and 2022, Fluck and Raveh-Rubin 2023a and b, Merdji et al., 2023. As a non-expert I'm wondering how the major claims presented here (i.e., lines 25 to 28 in the abstract) are related to the results of similar previous studies.
**RE:** Thanks for pointing these new studies of Flaounas et al., 2015 and 2022, Fluck and Raveh-Rubin 2023a and b, Merdji et al., 2023.

The study of Flaounas et al.(2015) is focused on regular dust events in the Mediterranean, not on extreme dust events, whereas Flaounas et al. (2022) did a study on cycles on the Mediterranean (with a rather short reference to dust). Nonetheless the comparison with the scenarios we observe is interesting, since the anticyclonic blocking over Iberia that we observe during extreme dust events hinder the entry of cyclones from the Atlantic to the Mediterranean.

The study of Fluck and Raveh-Rubin (2023a) is also focused on regular dust events over the Mediterranean, which do not occur under the same meteorological scenario. The comparison with our results is also of interest.

The study of Fluck and Raveh-Rubin (2023b) is very interesting and useful for us. This study is focused on 4 extreme dust events in North Africa; it clearly shows that extremely high $PM_{10}$ ($800 - 1220\ \mu g/m^3$) and ($PM_{2.5}$: $600 - 1230\ \mu g/m^3$) concentrations linked to dust episodes are characteristic of regions locate in the southern Sahara (e.g. Mauritania and Niger) and not in regions as the Canary Islands and mainland Spain. This study also shows other interesting result, the extreme dust events in Southern Sahara are linked to H-L dipoles, i.e. a high - anticyclone H located over Atlantic and North Africa and a cyclone - L over the Mediterranean, a meteorological configuration that results in strong southern winds. In our study we found the extreme dust events are linked to the northward winds linked to L–H dipoles. This comparison will be included in the revised version of the manuscript.

Merdji et al. (2023) preset a dust climatology, based on remote sensing (CALIPSO and MODIS), of regular dust events, no extreme dust events. This study present interesting results, showing that Saharan dust over Europe and the Canary Islands mainly occur in summer, a fact that also highlight the anomaly of the extreme dust events we report. This will also be included in the revised version of the manuscript.

References:

Flaounas, E., Kotroni, V., Lagouvardos, K., Kazadzis, S., Gkikas, A., & Hatzianastassiou, N. (2015). Cyclone contribution to dust transport over the Mediterranean region. Atmospheric Science Letters, 16(4), 473–478. https://doi.org/10.1002/asl.584

Flaounas, E., Davolio, S., Raveh-Rubin, S., Pantillon, F., Miglietta, M. M., Gaertner, M. A., Hatzaki, M., Homar, V., Khodayar, S., Korres, G., Kotroni, V., Kushta, J., Reale, M., & Ricard, D. (2022). Mediterranean cyclones: Current knowledge and open questions on dynamics, prediction, climatology and impacts. Weather and Climate Dynamics, 3(1), 173–208. https://doi.org/10.5194/wcd-3-173-2022

Fluck, E., & Raveh-Rubin, S. (2023a). A 16-year climatology of the link between dry air intrusions and large-scale dust storms in North Africa. Atmospheric Research, 292, 106844. https://doi.org/10.1016/j.atmosres.2023.106844

Fluck, E., & Raveh-Rubin, S. (2023b). Dry air intrusions link Rossby wave breaking to large-scale dust storms in Northwest Africa: Four extreme cases. Atmospheric Research, 286, 106663. https://doi.org/10.1016/j.atmosres.2023.106663

Merdji, A. B., Lu, C., Xu, X., & Mhawish, A. (2023). Long-term three-dimensional distribution and transport of Saharan dust: Observation from CALIPSO, MODIS, and reanalysis data. Atmospheric Research, 286, 106658. https://doi.org/10.1016/j.atmosres.2023.106658

**R2C3**
The authors might want to comment on the seasonal aspect of their results. Events with northward transport of Saharan Dust are typically more frequent during spring and summer. However, the events described here occurred in the winter and early spring.
**RE:** this is an interesting observation which have been included in the revised version of the manuscript. Thanks!

**R2C4**
Given that only Spanish observations are reported I think the 'Europe' should be replaced by 'Spain' in the title.
**RE:** the revised version of the manuscript includes, by suggestion of the reviewer 1, PMx data from Portugal.

**R2C5**
A comment about presentation style: the authors have amassed an impressively wide array of detailed information. It is generally well-presented and thus possible to follow. However, I confess that I struggled to comprehend all of the detail. The authors might consider trying to improve readability by removing some of the finer level detail and/or presenting it in a more concise format. For example, there is a lot of listing of PM10 and PM2.5 concentrations recorded at different stations, like the paragraph on lines 12-31 on page 19. This paragraph is very difficult to read. Is it necessary? One can already see the spatial distributions of daily averages in Fig. 9.
**RE:** thanks for this comment which definitively contributes to improve the readability. We have summarized that paragraph and transferee part of it to the supplement.

**R2C6**
Similarly, the authors use many compound plots combine many different elements and tend to be very detailed. Everything is explained well so this is not necessarily a problem. However, the authors may want to consider simplifying some of the figures considering the figures will be shrunk down in the final publication. For example, Fig. 1 currently covers 2 A4 pages simply with pictures and newspaper headlines. There is a lot of redundant information and many details will anyway be lost when the figure is shrunken down.
**RE:** as the reviewer points we did an effort to provide all needed information of a huge amounts of data in a few (as low as possible) plots. To merge Fig1A and 1B plots would result in very small pictures, actually difficult to see. These are historic events, so we have tried to properly report, in detail, them. Thanks for the comment.

**R2C7**
The authors introduce a new term: duxt events. I suggest they provide a more precise definition of this term and what differentiates duxt and regular dust events.
**RE:** we use the term duxt to refer extreme dust events, i.e. an anomalous high dust concentrations, much higher than dust concentrations during the regular intense events, we worked with events in which (1h or 30 min average) PM10 concentrations are > 1000 ug/m3. Probably, it would be ideal that international organizations, as the World Meteorological Organization, within the frame of the Sand and Dust Storm Warning Advisory and Assessment System (SDS-WAS), considered to create a definition, as already done for heat waves.

**R2C8**
P2, L4: Typo. 'Mayor' instead of 'Major'
**RE:** thanks for pointing this, correction included in the revised version of the manuscript.

**R2C9**
P2, L12: Typo...'loos'.
**RE:** thanks, corrected.

**R2C10**

P2, L13: Typo...'loose'

**RE:** it is as loose. Thanks

**R2C11**

P2, L20: The English grammar in the last part of this sentence is wrong, could be changed to "..., while dust events with PM10 > 100 ug/m3 are unusual."

**RE:** thanks for this correction, which has been included in the final version of the manuscript.

**R2C12**

P4, L8: Typo...'abord'.

**RE:** corrected. Thanks!

**R2C13**

P7, L15: Typo...'loose'.

**RE:** corrected (as loss). Thanks!

**R2C14**

P8, L2: The difference in PM2.5/PM10 ratios between duxt and regular dust events is somewhat surprising and possibly deserves further comment. The implication is that the size distribution of dust particles is shifted towards larger diameters during the duxt events. It could be interesting to speculate why this is the case. To validate the result and prove it is not an artifact of the interpolation method, it would be interesting to know if the difference in ratios between the duxt and regular dust events also occurs at the 4 AQMSs capable of recording PM10 values > 1000 ug/m3.

**RE:** the reviewer is wright; the particle size distribution is shifted to coarser particles during duxt events. We are now working in a new study of extreme dust events based on measured particle size distribution from 10 nanometers to 20 microns (measured with a Scanning Mobility Particle Sizer-SMPS- and an Aerodynamic Particle Sizers-APS-), these data confirm that size distribution during duxt events is shift to coarser diameters. The PM2.5/PM10 measured by the monitors can be observed before and after saturation (without any possibility of artifact) in Fig. 3B (for the case of El Charco site, Fuerteventura), where it can be see that it is of about 0.20; the same behavior is observed in the other sites. The proposal of the reviewer is very interesting, we have checked the data and found that during duxt events PM2.5/PM10 ratios measured with monitors that did not experienced saturation was 0.19 in Mercado Central (Las Palmas De Gran Canaria), 0.22 in Tome Cano (Santa Cruz De Tenerife) and 0.17 in Puerto Del Coto (Madrid), i.e. ratios lower than during regular dust events.

**R2C15**

P9, Fig. 1: The number of points marked invalid in panels A) and B) do not match.

**RE:** corrected. Thanks.

**R2C16**

P10, L19 and L22: I suggest writing 'and' explicitly rather than '+' to avoid confusion.

**RE:** modified as suggested by the reviewer. Thanks

END OF REPORT Author's response